# Rethinking supervised pre-training for better downstream transferring

**Yutong Feng** *
BNRist, THUIBCS, KLISS, BLBCI
School of Software, Tsinghua University
fyt19@mails.tsinghua.edu.cn

**Jianwen Jiang** *
Alibaba Group
jianwen.jjw@alibaba-inc.com

**Mingqian Tang**
Alibaba Group
mingqian.tmq@alibaba-inc.com

**Rong Jin**
Alibaba Group
jinrong.jr@alibaba-inc.com

**Yue Gao** †
BNRist, THUIBCS, KLISS, BLBCI
School of Software, Tsinghua University
gaoyue@tsinghua.edu.cn

## Abstract

The pretrain-finetune paradigm has shown outstanding performance on many applications of deep learning, where a model is pre-trained on an upstream large dataset (*e.g.* ImageNet), and is then fine-tuned to different downstream tasks. Though for most cases, the pre-training stage is conducted based on supervised methods, recent works on self-supervised pre-training have shown powerful transferability and even outperform supervised pre-training on multiple downstream tasks. It thus remains as an open question how to better generalize supervised pre-training model to downstream tasks. In this paper, we argue that the worse transferability of existing supervised pre-training methods arise from the negligence of valuable intra-class semantic difference. This is because these methods tend to push images from the same class close to each other despite of the large diversity in their visual contents, a problem to which referred as "overfit of upstream tasks". To alleviate this problem, we propose a new supervised pre-training method based on Leave-One-Out K-Nearest-Neighbor, or LOOK for short. It relieves the problem of overfitting upstream tasks by only requiring each image to share its class label with most of its $k$ nearest neighbors, thus allowing each class to exhibit a multi-mode distribution and consequentially preserving part of intra-class difference for better transferring to downstream tasks. We developed efficient implementation of the proposed method that scales well to large datasets. Extensive empirical studies on multiple downstream tasks show that LOOK outperforms other state-of-the-art methods for supervised and self-supervised pre-training.

## 1 Introduction

Pre-training neural networks on upstream datasets and fine-tuneing the pre-trained model on downstream tasks has been an important methodology in applications of deep learning (Tan et al., 2018). Such a **pretrain-finetune** paradigm generally works with pre-training on large-scale diverse datasets and fine-tuning on small specific datasets, and has been widely applied in a number of applications (Devlin et al., 2019; Kolesnikov et al., 2020; Brown et al., 2020). Specifically, in the area of computer vision, we often apply supervised learning methods (*e.g.* cross entropy) to pre-train a model from labeled dataset (*e.g.* ImageNet (Deng et al., 2009) and Kinetics (Carreira & Zisserman, 2017)),

---

*Equal contributions. This work was done when Yutong Feng was intern at Alibaba Group.
†Corresponding author.

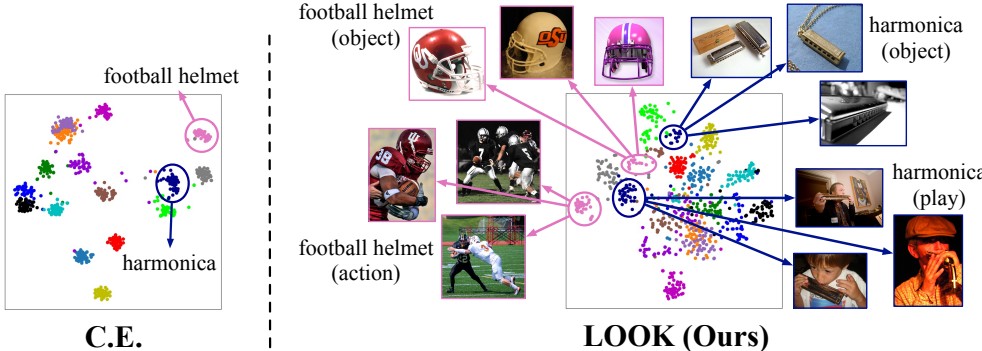

Figure 1: **Visualization of feature distribution pre-trained by C.E. and LOOK on ImageNet.** Taking the "football helmet" category as example, C.E. pushes all the samples into one cluster. However, there exist potential sub-categories representing the helmet iteself and its usage in football match, respectively. The "harmonica" class shows similar cases. Pushing samples from the two sub-category with completely different appearance will damage of representation learning process for downstream transferring. While our LOOK pre-trained model could adaptively separate them into different clusters, and thus preserve more valuable semantic features for better transferring.

and fine-tune it on downstream tasks such as object detection (Ren et al., 2017), instance segmentation (He et al., 2017), and video understanding (Gu et al., 2018).

Except supervised pre-training, recent works demonstrate that self-supervised pre-training without label information can also learn effective representation from upstream data and even surpass supervised methods when transferring to downstream tasks (Chen et al., 2020a; He et al., 2020; Grill et al., 2020; Chen & He, 2021; Zbontar et al., 2021). Unlike supervised pre-training that focuses on category-level discrimination, self-supervised pre-training is mainly based on instance discrimination, where models are trained to keep each instance and its augmentations close to each other, and at the same time, separate them from other instances and their augmentations. It effectively captures many important and discriminative features that are useful for downstream tasks. However, without appropriate guidance of supervision information, self-supervised pre-training lacks the ability of mining high-level semantic features and may capture detailed but irrelevant features (e.g. visual features related to special background), resulting in unsatisfying performance on challenging downstream tasks, *e.g.* fine-grained classification (Islam et al., 2021).

Therefore, we aim to improve the downstream transferability for supervised pre-training. Figure 2 illustrates two representative supervised pre-training methods, cross entropy (C.E.) and supervised contrastive learning (SupCon, Khosla et al. (2020)), a soft-nearest neighbors loss (Goldberger et al., 2005; Wu et al., 2018a). To distinguish instances from different classes, they minimize intra-class variance by pushing all the instances of the same class close to each or to certain centers. For a class with diverse visual appearance (e.g. the examples from Figure 2), these approaches may "ruin" the natural representation of images by bringing images with completely different visual appearances next to each other. As a result, they tend to skip features that capture intra-class difference but are less correlated with classes defined in upstream tasks, leading to overfitting upstream tasks.

In this paper, we propose a new supervised pre-training method based on **Leave-One-Out K-nearest-neighbor** classification, or **LOOK** for short, that effectively alleviates the problem of neglecting intra-class difference and thus significantly improves transferability for downstream tasks. In particular, a weighted kNN classifier is used to replace the linear or MLP predictor in the last layer of deep neural network, and a leave-one-out classification error is used as the loss function for optimization. Because of the nature of kNN classifier, each instance is only required to share the same class with most of its k nearest neighbors, allowing each class to exhibit multi-mode distribution and consequentially better preserving the features related to intra-class difference, as shown in Figure 2. We also develop efficient implementation for LOOK that scales well to large datasets. Extensive empirical studies demonstrated that LOOK has better transferrability for downstream tasks than the existing methods for supervised and self-supervised pre-training.

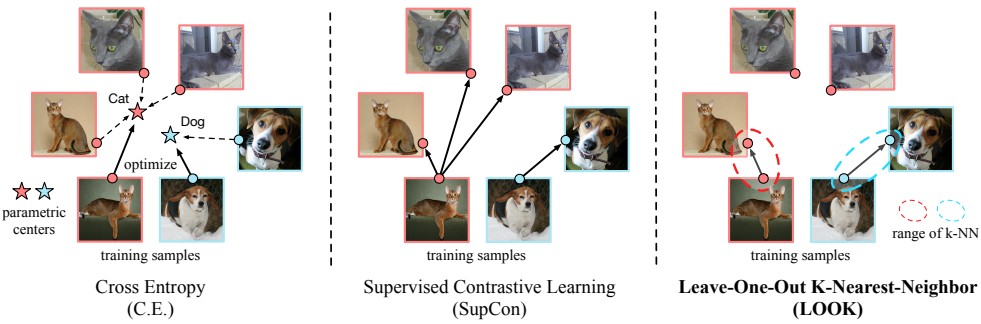

Figure 2: **Comparison of LOOK and existing supervised pre-training methods**. For C.E. and SupCon, they push all samples from the same class to certain centers or closer to each other, respectively, while LOOK only requires samples next to at most their $k$ nearest neighbors.

## 2 RELATED WORKS

Pre-training models have achieved remarkable performance on multiple fields of artificial intelligence. In computer vision, various DNN architectures (Krizhevsky et al., 2012; Simonyan & Zisserman, 2014; He et al., 2016; Dosovitskiy et al., 2020) show significant performance improvements on downstream tasks when pre-trained from large labeled datasets, *e.g.*, ImageNet (Deng et al., 2009), JFT-300M (Sun et al., 2017). Pre-trained model provide an essential initialization for complex vision tasks, such as detection, segmentation and *etc.*, which work as a important part for the convergence of the model training. In natural language processing, pre-trained models (Devlin et al., 2019; Liu et al., 2019; Brown et al., 2020) also have made great progress.

Recently, contrastive learning has made impressive progress in the representation learning task of computer vision. Self-supervised learning methods (Chen et al., 2020a; He et al., 2020) are employed to pre-train on ImageNet dataset and significantly improve the transferability on multiple downstream tasks over supervised counterparts (Islam et al., 2021). Derived from contrastive learning, more self-supervised learning methods (Chen et al., 2020b; Grill et al., 2020; Zbontar et al., 2021; Chen & He, 2021) also achieve impressive performance on downstream tasks. Unlike earlier studies of contrastive learning, these methods introduce either momentum encoder or additional projector and predictor to avoid the introduction of negative pairs, significantly improving the performance. Apart from self-supervised learning, contrastive learning is also developed in supervised pre-training (Khosla et al., 2020; Islam et al., 2021) and further boost the transferability of the pre-trained model. In addition to self-supervised methods based on contrastive learning, there are also some unsupervised pre-training methods that focus on learning representation features by designing clever pretext tasks (Noroozi & Favaro, 2016; Gidaris et al., 2018; Trinh et al., 2019; Bao et al., 2021) and structure formed clustering (Caron et al., 2018; Asano et al., 2020; Li et al., 2021).

Besides designing pre-training method, there are also works (Ganin & Lempitsky, 2015; Yosinski et al., 2014; Li et al., 2018; 2019; Chen et al., 2019a; Tan et al., 2018) focused on designing downstream training methods to improve transferablity through eliminating domain shifts between the upstream and downstream datasets or utilize an inductive bias to improve transferability.

## 3 LOOK: A LEAVE-ONE-OUT KNN BASED PRE-TRAINING METHOD

### 3.1 MOTIVATION OF LOO-KNN

As already discussed in the introduction, existing supervised pre-training methods suffer from the problem of overfitting upstream tasks, *i.e.* they tend to keep instances from the same class close to each other, thus neglecting intra-class difference that may be important for downstream tasks. We believe that source of this problem arises from simple prediction models (*e.g.* linear classifier with fully-connected layer) are used in supervised pre-training methods. Figure 1 shows the data distribution resulting from C.E. based pre-training. We see clearly that using a simple linear classifier essentially requires all data points of the same class close to the class center. As a remedy,

we propose to replace simple prediction model with kNN classifier. The kNN classifier is much more powerful than a linear prediction model. It can fit almost any decision boundary (Bishop, 2006). Since it makes prediction without training, kNN has also been used for monitoring model's convergence in the process of self-supervised learning (Wu et al., 2018b; Chen & He, 2021).

Compared to the linear prediction model, kNN is advantageous for pre-training because it does not require all the examples of the same class to form one tight cluster. Given one query sample, as long as the number of samples with the same label of query takes the majority of its kNN, the label of query could be correctly predicted. Thus, each sample only need to share the same class with most of its $k$ nearest neighbors, and the resulting distribution can be multi-mode, instead of single-mode. Figure 1 shows the data distributions from cross entropy (C.E.) based supervised pre-training and the proposed LOOK method, respectively. We can see clearly that for the distribution from LOOK, data points from the same class can be distributed over multiple clusters while, for C.E., data points from the same class tend to form one cluster. It is this nice property of kNN classifier that allows us to preserve part of intra-class difference, leading to better performance for downstream tasks.

## 3.2 ALGORITHM LOOK: MATHEMATICAL DESCRIPTION

As mentioned before, we will use kNN for prediction model, and to facilitate training, a leave-one-out loss is used for optimization. Below we provide detailed description. We start by formulating the pre-training process on visual datasets. Given a set of images $X = \{x_i\}_{i=1}^N$, with corresponding labels $Y = \{y_i\}_{i=1}^N$ when training in supervised scenarios, we aim at training a encoder $f_\theta(\cdot)$, *e.g.* CNN, to map each image $x_i$ into an embedding vector $z_i = f_\theta(x_i)$ in the high-dimension space. The trained parameters generally work as the initialization to fine-tune on downstream datasets.

Specifically, we use the weighted kNN for class prediction. Given sample $x_i$ and its embedding $z_i = f_\theta(x_i)$, suppose the kNN set of $z_i$ in the representation space is $\mathcal{N}_k(z_i)$, then we aggregate of its kNN labels based on the weights of pairwise distance, which is selected as the cosine distance:

$$\tilde{\mathbf{L}}_i = \Sigma_{z_j \in \mathcal{N}_k(z_i)} w_{i,j} \cdot \mathbb{1}_{y_j}, \quad w_{i,j} = \langle z_i, z_j \rangle = \frac{z_i \cdot z_j}{\| z_i \| \| z_j \|}, \tag{1}$$

where $\mathbb{1}_{y_j}$ is the one-hot vector in size of total classes $C$ with value 1 only at the position $y_j$ and value 0 at the remaining positions. Then we normalize the aggregated labels with softmax function and feed it into the negative log-likelihood function as the loss of LOOK for the $i$-th sample:

$$\mathcal{L}_{(x_i,y_i)}^{LOOK} = -log \frac{\exp(\tilde{\mathbf{L}}_{i,y_i}/\tau)}{\Sigma_{c=1}^C \exp(\tilde{\mathbf{L}}_{i,c}/\tau)} = -\log \frac{\Sigma_{z_j \in \mathcal{N}_k(z_i)} \exp\left(\langle z_i, z_j \rangle/\tau\right) \cdot 1_{y_j=y_i}}{\Sigma_{z_j \in \mathcal{N}_k(z_i)} \exp\left(\langle z_i, z_j \rangle\right)/\tau)}, \tag{2}$$

where $\tau$ is the temperature hyper-parameter of softmax to control the normalization process, and $1_{y_j=y_i}$ equals one when $y_j = y_i$ and otherwise zero.

Despite the simplicity, how to make LOOK work efficiently for large datasets remains a challenge. For large datasets, it is too time consuming to compute kNN for the entire dataset in an online fashion. We will then show our efficient implementation that makes LOOK scale to large datasets.

## 3.3 MAKING LOO-KNN SCALE TO LARGE DATASETS

The computational challenge of LOOK arises from two aspects. First, since the encoder is updated in an online fashion, we can't afford to update the representation vectors of every instance every time. As a result, we have to handle the discrepancy between the latest updated encoder and the encoder used to map instances into vectors. Second, due to the large data size, we can't afford to find the $k$ nearest neighbors by comparing each instance to the entire dataset. We thus have to approximate the entire dataset by a small number of profiles to make the computation efficient.

**Efficient construction of search space for kNN.** Since it is too time consuming to find $k$ nearest neighbors by comparing each instance to the entire database [1], we decide to construct a small search space $\mathcal{S} \subset \mathcal{D}$ to approximate the entire dataset $\mathcal{D}$ for kNN search. From our motivation, $\mathcal{S}$ should

---

[1] We are aware of efficient methods for $k$ nearest neighbors search, such as k-d tree (Chen et al., 2019b), LSH (Slaney & Casey, 2008), and quantization methods (Jegou et al., 2010). For large-scale training, the overheads of these methods can still be very significant, and we did not fully exploit this line of development in this work.

be large enough for the coverage of the whole dataset, and contain temporally synchronized embeddings that are generated from the training encoder at the same timestamp. Taking two extreme cases as example, we could maintain a memory bank as $S$ with same size of the dataset and continuously update embeddings of current training samples. The main problem with this approach is that to avoid updating the embedding vectors for the entire dataset whenever the encoder is updated, we have to generate non-synchronized embedding for the memory bank, leading to significant error in distance measurements. In contrast, a light-weight choice is to adapt the mini-batch as $S$ with completely synchronized embeddings from the same timestamp. However, since the mini-batch only contains a small number of samples from the entire dataset, leading to a large error in identifying $k$ nearest neighbors and consequentially poor classification compared to other supervised learning methods. To address this problem, we adapt the momentum queue mechanism from MoCo, a queue of size $q$ updated by first-in first-out (FIFO) strategy, containing embeddings from a momentum encoder. Its parameter $\theta_m$ is updated as $\theta_m^{t+1} = m \cdot \theta_m^t + (1 - m) \cdot \theta^t$, where $\theta_t$ is the parameter of the original encoder at time $t$, also known as "online encoder", and $m \in [0, 1]$ is the momentum parameter. The momentum encoder is updated much slower than the online encoder with larger $m$, *e.g.* 0.999, which helps us to maintain approximated synchronized embeddings in the queue with larger size.

**Predictor module for faster convergence.** Although using the momentum encoder helps the training process, it still pulls embeddings from the online encoder and those from momentum encoders close to each other, leading to slow convergence. To address this problem, we introduce the projector-predictor to alleviate the discrepancy between the online encoder and momentum encoder, similar to BYOL (Grill et al., 2020). More specifically, an additional MLP module $p(\cdot)$ is appended after the online encoder as the predictor module, and the output $p(z_i)$ is used for searching kNN based on the embedding sets generated from the momentum encoder. With such a module, we provide a buffer from the online encoder to the momentum encoder to achieve faster convergence.

**Dynamic adjustments of hyper-parameters for kNN.** The size $k$ and temperature $\tau$ decide the range of kNN label aggregation, and we observe it is important to adjust them along the training. At the beginning, samples are randomly distributed, requiring larger range of aggregation. When coming to the later stage of training, since multi-mode distribution has already formed, we require smaller range of aggregation to avoid pulling the multiple clusters together. Therefore, we utilize a decaying strategy that decreases $k$ throughout training to adjust aggregation under fixed $\tau$. Empirical studies shows similar performance of decaying of $\tau$ with larger enough $k$.

**Avoiding gradient explosion problem.** We observe gradient explosion at the very beginning of LOOK, which is caused by a cold-start problem that the kNN set $\mathcal{N}_k(z_i)$ contains no samples from the same class of $z_i$. To address this problem, we first fill the momentum queue without training to ensure the size of searching space. Then we apply extreme value filtering strategy for the LOOK output to be no less than a small value $\epsilon$ (*e.g.* $1e - 5$) to avoid gradient explosion.

## 4 EXPERIMENTS

### 4.1 EXPERIMENTAL SETTINGS

**Datasets.** For the upstream dataset, we use the ImageNet ILSVRC (Deng et al., 2009) with 1.28M images of 1K categories since most pre-training methods for comparison are trained on ImageNet. For the downstream datasets, we select 9 fine-grained datasets from varying domains to evaluate model's transferability inspired by Islam et al. (2021), including the Aircraft (Maji et al., 2013), Cars (Krause et al., 2013), DTD (Cimpoi et al., 2014), EuroSAT (Helber et al., 2019), Flowers (Nilsback & Zisserman, 2008), ISIC (Codella et al., 2019), Kaokore (Tian et al., 2020), Omniglot (Lake et al., 2015) and Pets (Patino et al., 2016). Dataset statistics are summarized in the appendix.

**Upstream Pre-training methods.** The compared pre-training methods are under supervised and self-supervised settings. For supervised methods, we reproduce or adapt the cross entropy (C.E.) guided training, supervised contrastive learning (SupCon, Khosla et al. (2020)) and the examplar-based supervised learning Examplar-v2 (Zhao et al., 2020). We also implement a SupCon+SSL version by adding additional MoCo loss to SupCon (Islam et al., 2021). For self-supervised methods, we compare recent representative works including SimCLR (Chen et al., 2020a), MoCo-v2 (Chen et al., 2020b), BYOL (Grill et al., 2020) and SimSiam (Chen & He, 2021). For the implementation of our proposed LOOK, we use queue size $q = 65536$, momentum $m = 0.99$, temperature $\tau = 1.0$

Table 1: **Downstream transferring results with linear fine-tuning.** For each method, "epochs" indicates their pre-training epochs and "aug++" indicates whether trained with strong data augmentation. † suggests that models are from official open-source codebases.

| method | epochs | aug++ | mean | Aircraft | Cars | DTD | EuroSAT | Flowers | ISIC | Kaokore | Omniglot | Pets |
|---|---|---|---|---|---|---|---|---|---|---|---|---|
| C.E. | 90 | | 68.46 | 37.08 | 46.85 | 68.30 | 94.62 | 89.59 | 73.84 | 78.08 | 37.71 | 90.05 |
| C.E. | 90 | ✓ | 67.62 | 40.98 | 46.54 | 67.29 | 91.77 | 87.56 | 69.98 | 74.01 | 41.00 | 89.48 |
| SupCon | 90 | ✓ | 63.29 | 34.32 | 38.91 | 65.96 | 90.38 | 81.09 | 68.89 | 70.81 | 31.34 | 87.90 |
| SupCon+SSL | 90 | ✓ | 71.17 | 44.52 | 52.57 | 70.16 | 94.67 | 90.37 | 73.68 | 76.48 | 48.99 | 89.13 |
| Examplar-v2 † | 200 | ✓ | 73.96 | 50.95 | 54.09 | 71.86 | 95.75 | 91.22 | 76.34 | 78.69 | 61.53 | 85.25 |
| SimCLR † | 1000 | ✓ | 69.55 | 45.96 | 49.78 | 67.02 | 94.14 | 88.40 | 72.71 | 79.42 | 51.04 | 77.50 |
| BYOL † | 1000 | ✓ | 74.71 | 50.50 | 61.47 | 71.54 | 94.98 | 93.41 | 75.67 | 79.42 | 58.07 | 87.33 |
| MoCo-v2 † | 200 | ✓ | 73.93 | 50.11 | 53.21 | 72.07 | 95.98 | 91.02 | 76.81 | 79.68 | 65.66 | 80.84 |
| SimSiam † | 100 | ✓ | 75.88 | 53.59 | 61.48 | 72.82 | 95.70 | 92.81 | 75.64 | 80.76 | 67.32 | 82.77 |
| **LOOK (Ours)** | 90 | | 77.60 | 56.83 | 69.20 | 71.22 | 95.81 | 94.94 | 76.51 | 79.19 | 64.57 | 90.11 |
| **LOOK (Ours)** | 90 | ✓ | **78.55** | 59.98 | 71.91 | 72.34 | 95.00 | 94.68 | 74.98 | 79.31 | 67.83 | 90.95 |

Table 2: **Downstream transferring results with fully fine-tuning.** See caption of Table 1 for detail.

| method | epochs | aug++ | mean | Aircraft | Cars | DTD | EuroSAT | Flowers | ISIC | Kaokore | Omniglot | Pets |
|---|---|---|---|---|---|---|---|---|---|---|---|---|
| C.E. | 90 | | 87.77 | 81.22 | 87.82 | 73.46 | 99.10 | 96.05 | 80.20 | 88.92 | 90.29 | 92.83 |
| C.E. | 90 | ✓ | 88.13 | 83.20 | 89.16 | 73.09 | 98.70 | 95.70 | 80.27 | 89.29 | 91.10 | 92.70 |
| SupCon | 90 | ✓ | 87.27 | 83.59 | 88.82 | 70.85 | 98.77 | 94.65 | 78.87 | 86.82 | 90.39 | 92.64 |
| SupCon+SSL | 90 | ✓ | 87.74 | 81.76 | 88.60 | 72.71 | 98.93 | 95.71 | 80.07 | 88.79 | 91.20 | 91.93 |
| Examplar-v2 † | 200 | ✓ | 88.72 | 84.28 | 89.44 | 74.63 | 99.00 | 96.08 | 82.20 | 89.16 | 92.64 | 91.03 |
| SimCLR † | 1000 | ✓ | 82.31 | 70.06 | 79.32 | 69.84 | 97.68 | 91.46 | 76.97 | 86.08 | 85.94 | 83.40 |
| BYOL † | 1000 | ✓ | 86.80 | 78.37 | 85.91 | 74.84 | 98.79 | 95.54 | 80.17 | 86.82 | 90.31 | 90.41 |
| MoCo-v2 † | 200 | ✓ | 88.61 | 85.11 | 90.29 | 75.00 | 98.90 | 96.04 | 81.17 | 89.66 | 91.89 | 89.47 |
| SimSiam † | 100 | ✓ | 87.95 | 86.35 | 90.50 | 71.65 | 99.10 | 95.74 | 76.44 | 89.29 | 93.55 | 88.96 |
| **LOOK (Ours)** | 90 | | 88.03 | 83.77 | 90.27 | 72.13 | 98.84 | 96.37 | 77.64 | 88.55 | 92.19 | 92.48 |
| **LOOK (Ours)** | 90 | ✓ | **88.79** | 85.54 | 90.70 | 72.93 | 98.81 | 96.57 | 80.47 | 89.04 | 92.74 | 92.31 |

and decaying $k$ linearly from 400 to 40. All the implemented methods are trained by 90 epochs with an initial learning rate of 0.1, multiplied by 0.1 for every 30 epochs. We use ResNet-50 (He et al., 2016) as the backbone encoder and train using SGD optimizer with momentum 0.9 and weight decay 0.0001. Since SupCon and all the self-supervised methods are trained with strong augmentation, we implement two versions of C.E. and LOOK with normal and strong data augmentation, respectively.

**Downstream Fine-tuning methods.** When evaluating the pre-trained models on downstream datasets, there are various fine-tuning methods to transfer models. In the main experiments, our first evaluations are based on two simple fine-tuning strategies: linear fine-tuning and fully fine-tuning. The linear fine-tuning fixes the parameters of encoder and only trains a new classifier module, while the fully fine-tuning trains the whole model. We further investigate to apply more advanced fine-tuning methods from recent studies of transfer learning. Since some of the pre-trained models are trained without linear classifier on upstream, we follow Islam et al. (2021) to append a batch-normalization layer without affine parameters after the encoder to generate properly distributed features for classification learning. During the fine-tuning stage, we train on the downstream datasets 50 epochs and decay the learning rate at the 25 and 37 epochs by 0.1. For the remaining hyper-parameters of training, we conduct grid search for the initial learning rate of 0.001, 0.01 and 0.1, weight decay of 0, $1e-4$ and $1e-5$, batch size of 32 and 128, and report the downstream performance with training on train and validation sets under the searched hyper-parameters.

## 4.2 Main Results of Downstream Transferring

**Linear fine-tuning.** The experimental results of linear fine-tuning are provided in Table 1, and we calculate the mean accuracy of the 9 downstream fine-grained datasets. We observe that LOOK outperforms all the compared methods. The existing supervised pre-training, *i.e.* C.E., SupCon and Examplar-v2, show worse transferability compared with self-supervised methods due to their upstream over-fitting problem, where the results of SupCon without SSL are even worse for its stronger requirement of pushing all samples of the same category into one cluster. In contrast, as a supervised pre-training method, LOOK significantly improves the transferring results via alleviating

Table 3: **Downstream transferring results of different fine-tuning methods.** The percentage under each dataset indicates the sampling rate of training samples.

| pre-train | fine-tune | Cars | | | | Aircraft | | | |
|---|---|---|---|---|---|---|---|---|---|
| | | 15% | 30% | 50% | 100% | 15% | 30% | 50% | 100% |
| C.E. | Baseline | 41.2 | 63.8 | 77.6 | 88.3 | 43.8 | 59.8 | 70.5 | **83.4** |
| LOOK (Ours) | | **46.8** | **69.1** | **80.6** | **89.8** | **48.1** | **64.1** | **72.4** | 83.2 |
| C.E. | BSS | 42.0 | 64.8 | 78.0 | 88.3 | 44.1 | 59.9 | 71.2 | 82.1 |
| LOOK (Ours) | (Chen et al., 2019a) | **47.4** | **69.7** | **81.3** | **89.7** | **48.7** | **63.6** | **72.7** | **82.9** |
| C.E. | DELTA | 39.9 | 64.0 | 78.0 | 88.7 | 43.0 | 60.4 | 70.3 | 82.5 |
| LOOK (Ours) | (Li et al., 2019) | **47.0** | **70.4** | **82.1** | **90.3** | **47.7** | **64.8** | **72.3** | **83.8** |
| C.E. | StochNorm | 41.1 | 65.0 | 77.8 | 88.4 | 43.4 | 60.3 | 70.0 | 82.0 |
| LOOK (Ours) | (Kou et al., 2020) | **47.7** | **69.2** | **80.1** | **89.6** | **48.4** | **63.8** | **71.9** | **82.6** |

Table 4: **Linear fine-tuning results of varying hyper-parameters in the LOOK pre-training**, including the queue size $q$, momentum $m$ and number of nearest neighbors $k$. Models are trained with the default settings that $q = 65536$, $m = 0.99$, $\tau = 1.0$ and decaying $k$ from 400 to 40.

| queue size | fine-tuning | momentum $m$ | fine-tuning | $k$ of kNN | fine-tuning |
|---|---|---|---|---|---|
| 65,536 | **78.55** | 0.9999 | 78.44 | 100 | 75.19 |
| 32,768 | 78.23 | 0.999 | 78.30 | 200 | **78.42** |
| 16,384 | 77.72 | 0.99 | **78.55** | 400 | 77.93 |
| 8,192 | 77.71 | 0.9 | 78.44 | 800 | 77.51 |

upstream over-fitting, with an improvement to C.E. with $10.9\%$ accuracy. Compared with self-supervised learning, LOOK also surpasses state of the art method, *i.e.* SimSiam, by $2.7\%$ mean accuracy via effectively leveraging the label information. It is also observed that though strong data augmentation boosts the self-supervised pre-training, it may introduce negative influence on supervised C.E. pre-training. Since the encoder for extracting features is frozen in linear fine-tuning, the experimental results indicate that LOOK could present more generalized representation based on pre-training, compared with existing supervised and self-supervised methods.

**Fully fine-tuning.** Table 2 shows the results of fully fine-tuning. It is worth noting that when coming to fully training of the entire model, the gap is reduced significantly. LOOK still achieves better mean accuracy compared with existing methods. Furthermore, we do observe, for a few datasets, that self-supervised learning indeed outperforms the proposed approach, indicating that we may need to combine the strength of LOOK with that of instance discrimination based methods.

**Advanced fine-tuning methods.** We further investigate the results with advanced fine-tuning methods in recent studies of transfer learning, including the BSS (Chen et al., 2019a), DELTA (Li et al., 2019) and StochNorm (Kou et al., 2020), and the naive fully fine-tuning is regarded as the baseline. Following the above methods, we also study the effect of using different sampling rate of downstream training samples based on Cars and Aircraft dataset. The results of C.E. and LOOK are listed in Table 3. We observe three advantages introduced by LOOK. First, compared to C.E., LOOK obtains remarkable improvements on almost all scenarios. Second, LOOK can collaborate and achieve consistency improvements with the advanced fine-tuning methods. Finally, with less training data, LOOK achieves larger gap of improvements, suggesting that generalization ability of LOOK learned representation can reduce the reliance on the amount of downstream training data.

## 4.3 ABLATION STUDIES

**On the configurations of LOOK.** We study the influence of hyper-parameters in LOOK on the downstream performance. From the first and second sub-table of Table 4, we show the results of varying queue size $q$ and momentum $m$ of the momentum queue. Experimental results suggest that LOOK show great robustness to the configuration of search space, which is crucial considering the evaluation of transferability is tough during the pre-training stage. The transferring results using

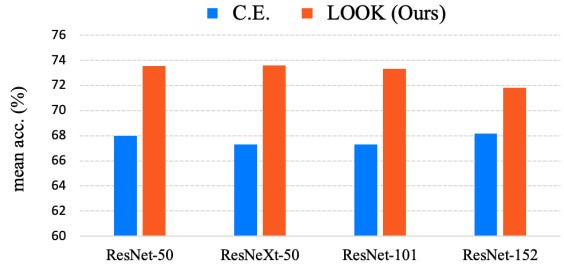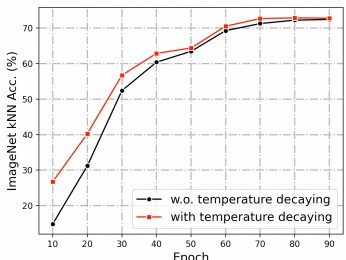

Figure 3: **Left:** Linear fine-tuning results using different types of backbone. **Right:** kNN monitoring accuracy during LOOK training with and without temperature decaying.

Table 5: **Results of memory-based fine-tuning, including voting and clustering.** The linear fine-tuning results of C.E. and LOOK are listed for reference.

| pre-train | finetune | # forward | mean | Aircraft | Cars | DTD | EuroSAT | Flowers | ISIC | Kaokore | Omniglot | Pets |
|-----------|----------|-----------|------|----------|------|-----|---------|---------|------|---------|----------|------|
| C.E. | linear ft. | 50 | 67.62 | 40.98 | 46.54 | 67.29 | 91.77 | 87.56 | 69.98 | 74.01 | 41.00 | 89.48 |
| LOOK | | | 78.55 | 59.98 | 71.91 | 72.34 | 95.00 | 94.68 | 74.98 | 79.31 | 67.83 | 90.95 |
| C.E. | clustering | 1 | 55.59 | 22.80 | 23.31 | 64.10 | 85.74 | 72.94 | 43.59 | 60.96 | 37.33 | 89.51 |
| LOOK | | | 64.72 | 39.60 | 36.21 | 66.38 | 90.33 | 87.28 | 52.21 | 69.46 | 52.33 | 88.72 |

different fixed $k$ of kNN are listed in the last sub-table of Table 4, which shows that when using fixed $k$, larger $k$ decreases the transferring performance while smaller $k$ may slow down the convergence of training. Besides $k$ decaying, we also monitor the kNN accuracy along the training to investigate the temperature decaying. Results in Figure 3 (right) shows that with a proper value of $\tau$ at the beginning time helps the model to converge faster compared with a constant temperature $\tau$.

**On the backbone model.** We investigate different types of encoder backbone to show the robustness of LOOK. For convenience, we compare with C.E. model from the torchvision codebase (Marcel & Rodriguez, 2010) with training only on downstream train set. Figure 3 (left) shows the linear fine-tuning results of C.E. and LOOK with the backbone of ResNet-50, ResNet-101, ResNet-152 and ResNeXt-50. The comparison suggests that LOOK could consistently outperform C.E. on varying encoder backbones and show robustness to its applied models as a pre-training method.

## 4.4 MEMORY-BASED FINE-TUNING WITHOUT TRAINING

In our main experiments, all the fine-tuning methods is conducted by optimizing a parametric classifier module for the downstream dataset. However, since the classification of LOOK on upstream is conducted in a non-parametric way based on the memory of embeddings and labels, we also explore strategies to follow the same way of non-parametric classification on downstream. Specifically, we aim at transferring to the downstream dataset via updating the embeddings and labels in the memory with the downstream samples. Throughout this way, we only need to forward the encoder function ONCE on the downstream dataset without additional training process. In practical, we first apply a layer-normalization layer without affine parameters after the encoder to directly normalize the output features. For better coverage of the downstream distribution, we propose to conduct clustering inside each category for generating better memory. The clustering is conducted on both the training and validation sets and we search the hyper-parameters including number of clusters, $k$ and temperature for the kNN classifier.

The results of memory-based fine-tuing are shown in Table 5, together with the linear fine-tuning results of C.E. and LOOK for reference. Though with only 1 time of forward on the downstream datasets, the proposed strategies based on LOOK pre-trained model achieve a comparable result with C.E. linear fine-tuning, which shows the generalization ability of LOOK pre-training and greatly reduces the computation cost of downstream transferring. Our exploration of transferring without training could lay foundations for faster and more convenient transferring in future works.

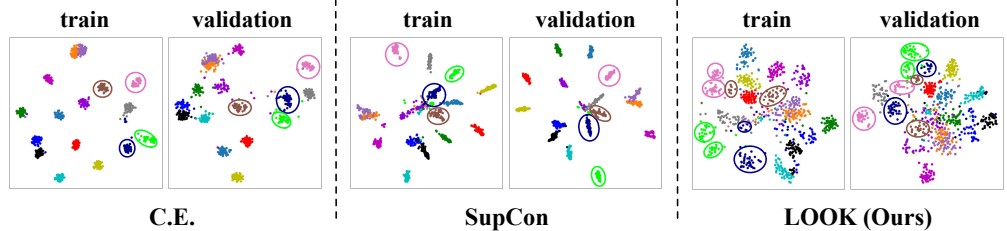

Figure 4: **Visualization of feature distribution on ImageNet using t-SNE.** We draw circles of obvious clusters with the same colors of corresponding categories.

Table 6: **Averaged intra and inter class distance of different pre-training methods,** where larger value indicates higher variance of intra-class or inter-class samples.

| method | C.E. | SupCon | MoCo-v2 | SimSiam | **LOOK (Ours)** |
|---|---|---|---|---|---|
| intra-class | $0.275 \pm 0.055$ | $0.328 \pm 0.123$ | $0.438 \pm 0.097$ | $0.398 \pm 0.053$ | $0.576 \pm 0.058$ |
| inter-class | $0.549 \pm 0.019$ | $0.819 \pm 0.021$ | $0.736 \pm 0.024$ | $0.574 \pm 0.019$ | $0.749 \pm 0.020$ |

## 4.5 WHY LOOK WORKS FOR BETTER TRANSFERRING?

To better understand the transferring and generalization ability of LOOK, we conduct deeper studies on its learned upstream representations. Specifically, we observe the representation distribution of LOOK and the compared methods in two way, *i.e.* a **qualitative** observation based on feature visualization and a **quantitative** observation based on similarity measurements.

**Feature Visualization.** Figure 4 visualizes the features of 10 random classes in the training and validation sets of ImageNet based on t-SNE (Van der Maaten & Hinton, 2008). Compared with visualization results of C.E. and SupCon, it is observed that the LOOK learned features could form multiple clustering distribution inside the same class, which is actually our basic motivation of proposing LOOK. With multiple clustering distribution, more available semantic information are preserved and is proved to provide more generalized features on downstream datasets.

**Measurements of intra-class and inter-class distance.** To further prove the distribution characteristics of LOOK, we follow Islam et al. (2021) to compute the intra-class and inter-class distance as a quantitative result of distribution. Table 6 shows the averaged $1 - cosine(\cdot, \cdot)$ measurement between samples of the same and different classes. C.E. and SupCon maintains smaller intra-class distance and larger inter-class distance. Self-supervised pre-training, MoCo-v2 and SimSiam, relax the intra-class tightness without label guidance, while their smaller inter-class distance compared with SupCon indicate that some high-level semantic information are missing to discriminate categories. LOOK achieves both larger intra-class and inter-class distance at the same time, which suggests that the learned representation generate clearer boundaries both inside and outside categories.

Based on the analysis of feature visualization and intra-class similarity measurement, we conclude that LOOK indeed adaptively learns multi-mode distribution inside categories and preserve more intra-class semantic features for better downstream transferring and generalization.

## 5 CONCLUSION

In this paper, we propose a new supervised pre-training method based on Leave-One-Out k-Nearest-Neighbor (LOOK) classifier for better downstream transferring. Compared with self-supervised pre-training, LOOK efficiently leverages the label information, and at the same time alleviate the problem of upstream over-fitting in existing supervised pre-training methods, which ignores intra-class difference semantic features that are valuable for transferring. We conduct extensive experiments on a number of downstream tasks. The experimental results show the superior performance of LOO-kNN against the SoTA supervised and self-supervised pre-training methods. Future works may explore strategies of efficiently combining LOOK and self-supervised methods to train powerful models with better generalized representation for downstream transferring.

ACKNOWLEDGMENTS

This work was supported by National Natural Science Funds of China (No. 62088102, 62021002, U1701262, 61671267). This work was supported by Alibaba Group through Alibaba Innovative Research Program.

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

# A  DATASETS

We present dataset statistics of our downstream transferring task in Table 7, which contains various types of technical, texture, satellite, natural, medical, illustrative, symbolic and natural visual contents and corresponding categories. For the train/validation/test split of each dataset, we follow the original split for those with official split file, *i.e.* Aircraft, DTD (the first official split), Flowers and Kaokore. For the remaining datasets with only train/test split, we preserve the test set, and randomly split the training set into training and validation sets with the proportion of 7 : 3 inside each category.

Table 7: Statistics of downstream tranferring datasets, including the train/validation/test split, number of classes and type of visual contents

| Dataset | # train | # val | # test | # classes | type |
|---|---|---|---|---|---|
| Aircraft (Maji et al., 2013) | 3,334 | 3,333 | 3,333 | 100 | technical |
| Cars (Krause et al., 2013) | 5,700 | 2,444 | 8,401 | 196 | technical |
| DTD (Cimpoi et al., 2014) | 1,880 | 1,880 | 1,880 | 47 | texture |
| EuroSAT (Helber et al., 2019) | 13,500 | 5,400 | 8,100 | 10 | satellite |
| Flowers (Nilsback & Zisserman, 2008) | 1,020 | 1,020 | 6,149 | 102 | natural |
| ISIC (Codella et al., 2019) | 5,007 | 2,003 | 3,005 | 7 | medical |
| Kaokore (Tian et al., 2020) | 6,568 | 826 | 821 | 8 | illustrative |
| Omniglot (Lake et al., 2015) | 6,590 | 2,636 | 3,954 | 1,623 | symbolic |
| Pets (Patino et al., 2016) | 2,575 | 1,105 | 3,669 | 37 | natural |

# B  DETAILS OF PRE-TRAINING AND FINE-TUNING METHODS

## B.1  SUPERVISED PRE-TRAINING.

For the proposed LOOK, to achieve a convenient fashion of "leave-one-out" that filtering out the query sample from the memory queue, we implement as follows. Given each mini-batch as input, we compute the LOOK loss before updating the queue with current mini-batch samples. Since the update of queue follows the "First-In-First-Out" (FIFO) strategy and queue size is significantly smaller than the dataset size, it has been a long period since the last time that current training samples are pushed into the queue. Thus, the mini-batch samples are very possible to be popped from the queue in earlier training iterations. Such a strategy is convenient for avoiding additional operations to filter out each training sample individually, which achieves satisfying performance in our experiments.

For all the compared pre-training methods, we present the implementation details or download source of pre-trained models.

**Cross Entropy (C.E.).** The most commonly used C.E. pre-training using a linear or MLP classifier guided by the cross entropy loss function. We implement two version of C.E. with and without strong data augmentation.

**Supervised Contrastive Learning** (SupCon, Khosla et al. (2020)). SupCon is supervised version of existing self-supervised contrastive learning. Besides the augmentation views of one training sample are regarded as "positive" samples, the other samples with the same label are also regarded as positive ones. SupCon requires training with large batch-size or MoCo trick. Since there is no open-source models with proper training epochs for comparison, we reproduce a 90 epochs pre-trained model with batch-size of 256 based on the MoCo trick, with queue size $8, 192$, momentum $0.999$ and temperature $0.07$ (Khosla et al., 2020), which is reported to show stronger performance the large batch-size training. We also follow Islam et al. (2021) to implement a SupCon+SSL that shows better transferablity, where the SSL method is selected as MoCo-v2.

**Exemplar-v2** (Zhao et al., 2020), which proposed a similar strategy to improve the existing contrastive learning that incorrectly pushing samples with the same label. We directly utilize the pre-trained model from its official open-source codebase.

**Difference with other supervised method.** In related works including Neighborhood Components Analysis (NCA), SNCA (mini-batch version of NCA), Examplar-v2 and SupCon, each sample will

be pulled togather with all samples in the dataset or memory bank with the same class. In contrast, the proposed LOOK only pull positive samples within the kNN, which alleviates the problem of neglecting intra-class difference and thus significantly improves transferability for downstream tasks. The difference could be observed from the positive set in the loss function, where all the samples are involved in NCA/SupCon and only samples within kNN are involved in LOOK. In other words, kNN is used to filter and select postive samples for constrastive learning in LOOK.

## B.2 SELF-SUPERVISED PRE-TRAINING.

For MoCo-v2 and SimSiam, we download their official provided pre-trained model based on 200 and 100 epochs of pre-training, respectively. For SimCLR and BYOL, we convert their official model weights from TensorFlow into PyTorch format, which are all trained for 1,000 epochs.

## B.3 FINE-TUNING METHODS

For the linear and fully fine-tuning, we conduct hyper-parameters searching based on the train and validation sets, and report the performance on test set based on optimization on both the train and validation sets with best hyper-parameters. In our main experiments, we also compare the performance of pre-trained models with advanced fine-tuning methods on downstream datasets. including BSS (Chen et al., 2019a), DELTA (Li et al., 2019) and StochNorm (Kou et al., 2020). The implementation of these fine-tuning methods are based on the open-source transfer learning codebase "Transfer-Learning-Library" (https://github.com/thuml/Transfer-Learning-Library). We also follow the split and subset sampling of the Aircraft and Cars datasets provided in this library. In detail, all the parameters of the model are available for update, similar to the full fine-tuning baseline. A similar configuration of linear and fully fine-tuning are adapted with total epochs of 50 (decaying at epoch 25 and 37, initial learning rate of 0.001 and batch size of 32. It is noted that the official split of the codebase is the train/val split, and we shows the best validation accuracy throughout training.

## C ADDITIONAL ANALYSIS INSIDE LOOK DURING TRAINING

### C.1 ON THE POSITIVE SAMPLES FALLING IN KNN

During the training stage of LOOK, the optimization is based on the kNN set of each sample, including the "positive" samples from the same class and "negative" samples from different classes. From our basic motivation, for those classes with higher intra-class variance, there should exist less positive samples inside their kNN sets, which will then be pushed into a sub-cluster of the whole class. While for those classes with similar appearance and semantics, there will exist more positive samples leading to tighter distribution of the class.

To investigate the composition of kNN, we calculate the number of samples in the memory queue for each class, and the averaged number of positive samples falling in the kNN. The results are shown in Figure 5. We refer to the ratio of positive samples falling in kNN as "falling ratio". Our observation and conclusion for the falling ratio are as follows:

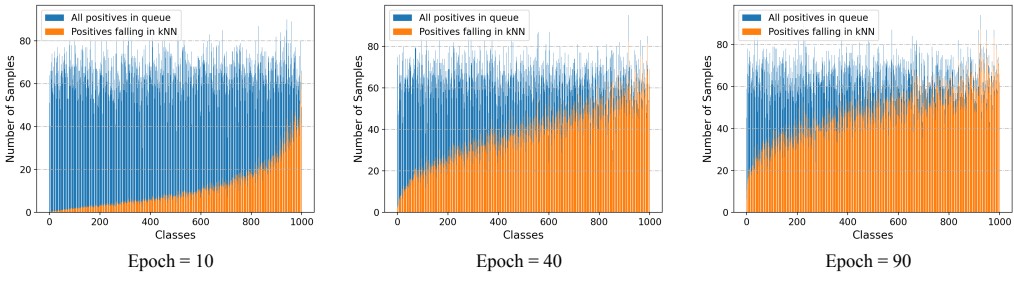

Figure 5: Number of positive samples in the memory queue and falling in the kNN during training on ImageNet, sorted by the ratio for all the classes.

| Category | Ratio | Samples |
|----------|-------|---------|
| yellow lady-slipper | 1.00 | |
| echidna | 0.98 | |
| sombrero | 0.63 | |
| cricket | 0.57 | |
| paperknife | 0.18 | |
| velvet | 0.14 | |

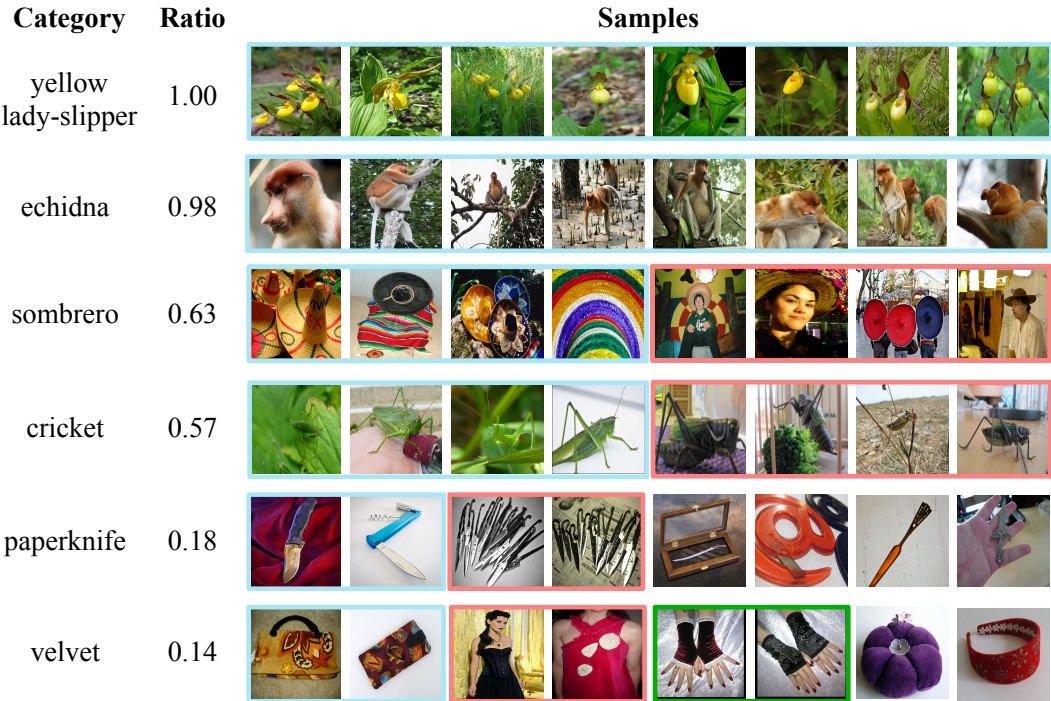

Figure 6: Visualization of randomly selected samples from classes with varying falling ratio, where we choose the top-2, bottom-2, and 2 middle classes as examples. We put similar samples of the same class together with colored bounding boxes for better observation.

**Compared among different classes**, an unbalanced distribution of the falling ratio is observed, which indicates that their intra-class variances are different. To further show the relation between the falling ratio and intra-class variance, we visualize randomly selected samples from several classes with different falling ratios in Figure 6. The top classes with almost all positive samples falling in kNN show very similar visual appearance among each other. While for classes with falling ratio of about 0.5, we could observe some obvious sub-classes, *e.g.* the yellow and green crickets. And for classes with lower falling ratio, the samples show more diversity, such as velvet used in different things (bag, dress, gloves, *etc.*). In conclusion, the falling ratio of LOOK could reflect different intra-class variance for learning efficient representation.

**Compared among different training stages**, for the earlier stage from epoch 10 to 40, an obvious increasing of the falling ratio is observed, where we use hyper-parameters with larger aggregation range to pull samples into the kNN in a sparse distributed representation space. While for the latter stage from epoch 40 to 90, the falling ratio increases much slower in spite of the continuously increasing accuracy of training (as shown in Figure 3). Such a phenomenon indicates that based on LOOK's relaxed restriction of kNN learning, improving the classification performance will not force all the positive samples into the kNN. Therefore, we could maintain the sub-clusters discovered during training.

## C.2 ON THE RANK OF POSITIVE AND NEGATIVE SAMPLES

From the observation in Figure 5, there will also remain some negative samples in the kNN, especially for those with lower falling ratio, where some negatives rank closer to the query sample than some positives. To better understand such cases, we illustrate one of them in Figure 7. As we have discussed in Figure 1, LOOK discovers two sub-classes for the class "football helmet" as the helmet itself and its usage in match, respectively. For the shown query image in training, we observe some negative samples ranked before the positive one, where the positive belongs to different sub-class from the query's and the negatives actually show more similar appearance to the query.

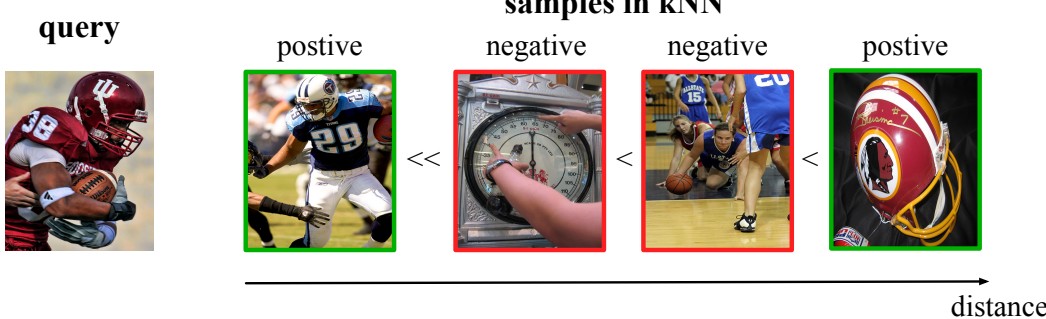

Figure 7: Illustration of cases when negative samples falling closer to the query image compared with some positive ones.

Based on the illustration in Figure 7, we conclude that for classes with higher intra-class variance, there will be a lower falling ratio and more negative samples of kNN. However, such a case is not bad because these negatives with similar appearance could occupy the left space of kNN and help to avoid positives from different sub-classes falling into the center of kNN.

### C.3 ON THE ABILITY OF DISCOVERING SUB-CLASSES

Based on analysis from the above subsections, the classes show different intra-class variance and number of potential sub-classes, and the number of positive samples maintained in the memory bank is highly related to ability of discovering sub-classes. It is noted that the motivation of LOOK is not to directly discover more sub-classes with clear boundaries, but to preserve more semantics by avoiding pulling samples together with completely different appearance. In spite of that, in this subsection, we take a further study to the effect of memory size on the ability of sub-class discovery, which will be conducted from two perspectives, *i.e.* empirical study and probability estimation. The following study is based on the assumption that each class could be uniformly divided into several sub-classes, which only serves as a simple analysis and could be further discussed in future work.

**Empirical Study.** Based on the expected number of samples for each sub-class in the queue, we could give a empirically estimated number of sub-classes that could be effectively learned based on experimental results. We refer to a similar work, SupCon (Khosla et al., 2020), that utilizes similar contrastive learning inside large mini-batch or memory bank for classification learning. From the Figure 4 of SupCon, the model could reach satisfying performance on ImageNet-1K with more than $2,000$ samples in batch, which contains $0.15\%$ of all the $1,300$ images per category. Based on the $5\%$ sampling rate of LOOK's default settings, empirically, there is a potential of capturing $\frac{5\%}{0.15\%} \approx 33$ sub-classes for the proposed method with the default queue size of $65,536$. It is noted that different from parametric classification learning that representing classes with parameters, LOOK adapts the non-parametric contrastive learning that continuously and rapidly update the memory samples throughout the dataset, which leaves sufficient times for each sample to meet and interact with samples from the same class or sub-class.

**Probability Estimation.** We further calculate the probability that all the sub-classes are sampled with at least one image in the queue based on the principle of tolerance and exclusion, *i.e.* $P(c) = \Sigma_{k=0}^{c}(-1)^k \frac{C(q(1-(k/c)),n)}{C(q,n)} k^c$, where $c$ is the number of sub-classes, $q$ is the queue size and $n$ is the sampling size. Results in Table 8 show that for less than 12 sub-classes, all the training samples are almost guaranteed to find samples in kNN with the same sub-class. It is noted that for contrastive-based methods, even one positive sample could serve for effective training, *e.g.* the another augmented view in self-supervised learning.

**Number of sub-classes in real-world dataset.** Based on the above analysis, we show the ability of LOOK under current settings to learn some sub-classes on ImageNet, which is consistent with the visualization results showing obvious sub-clusters. A further problem is what is the actual number of sub-classes in the real-world datasets. Since the definition of a "class" or "sub-class" is not fixed

Table 8: Probability that all the $c$ sub-classes are sampled with at least one image under the default setting of queue size of $65,536$.

| $c$ | $< 6$ | 8 | 12 | 16 | 20 | 24 | 28 | 32 |
|---|---|---|---|---|---|---|---|---|
| $\lfloor P(c) \rfloor$ | 0.9999 | 0.9991 | 0.9677 | 0.8252 | 0.6002 | 0.4088 | 0.2821 | 0.2026 |

based on the granularity and focused semantics, it is an open question to count all the sub-classes given a set of samples without artificial rules. A rough estimation to the averaged number of sub-classes discovered by LOOK could be the reciprocal of the averaged falling ratio, *i.e.* $\frac{1}{0.714} \approx 1.4$. We further analyze the relation of LOOK's components to the sub-classes discovery as follows:

- **Granularity:** The granularity of discovered sub-classes is mainly affected by two factors, *i.e.* the memory size and $k$ of LOOK. With larger memory bank storing enough samples to cover the whole class together with smaller $k$ to strictly control the neighbors of samples, we could reach more fine-grained sub-classes.

- **Semantics:** The rule of splitting sub-classes is based on the learned visual semantics, which is guided by the supervision of coarse labels. Along the training stage, the semantics captured and focused by the model determine the kNN structure for sub-class discovery.

**Future works related to sub-class discovery.** Since the memory size is positively related to the number of sub-classes, it is still an challenging problem how to efficiently utilize the memory space to model more intra-class variance. For the future works, it is important to explore how to adaptively arrange memories for classes with more sub-classes and maintain representative samples in the memory for better coverage of all sub-classes.

## D  UPSTREAM V.S. DOWNSTREAM PERFORMANCE

Though in this paper we focus on improving the downstream transferring performance of pre-trained models, in this section, we show the upstream performance for more comprehensive study of the proposed method. In Table 9, we show the upstream accuracy on ImageNet of the compared methods, together with the downstream performance for reference. For parametric classification methods or methods have been fine-tuned with linear classifier, we report the linear accuracy, while for the remaining we report the kNN accuracy ($k = 200$ and temperature $\tau = 0.1$).

Comparing the upstream and downstream results in Table 9, we observe that the downstream performance of one pre-training method is not highly related to its upstream performance. As we have discussed in the motivation of LOOK, methods with better upstream performance may fall in the over-fitting to the upstream datasets, leading to worse transferability on downstream tasks.

Table 9: Upstream accuracy on ImageNet (kNN and linear classifier) and downstream performance on 9 fine-grained datasets (linear or fully fine-tuning).

| method | aug++ | epochs | upstream (knn) | upstream (linear) | downstream (linear) | downstream (fully) |
|---|---|---|---|---|---|---|
| C.E. | | 90 | - | **75.5** | 65.2 | 84.3 |
| SupCon+SSL | ✓ | 90 | 72.8 | - | 67.8 | 83.9 |
| Examplar-v2 | ✓ | 200 | - | 68.9 | 69.6 | 84.9 |
| SimCLR | ✓ | 200 | - | 61.6 | 63.9 | 82.3 |
| MoCo-v2 | ✓ | 200 | - | 67.7 | 69.7 | 84.7 |
| BYOL | ✓ | 300 | - | 72.4 | 70.2 | 82.5 |
| SimSiam | ✓ | 100 | - | 68.3 | 71.7 | 84.3 |
| LOOK (Ours) | | 90 | 73.2 | - | 73.5 | **85.1** |
| LOOK (Ours) | ✓ | 90 | 72.8 | - | **74.1** | **85.1** |

Table 10: **Transferring results of objection detection and instance segmentation on PASCAL VOC and COCO.** "2V" indicates training with two augmented views of each image. All the compared methods are fine-tuned with the $1\times$ schedule.

| pre-train | epochs | 2V | VOC 07+12 detection | | | COCO detection | | | COCO instance seg. | | |
|---|---|---|---|---|---|---|---|---|---|---|---|
| | | | $AP^{bb}$ | $AP^{bb}_{50}$ | $AP^{bb}_{75}$ | $AP^{bb}$ | $AP^{bb}_{50}$ | $AP^{bb}_{75}$ | $AP^{mk}$ | $AP^{mk}_{50}$ | $AP^{mk}_{75}$ |
| scratch | - | | 33.8 | 60.2 | 33.1 | 26.4 | 44.0 | 27.8 | 29.3 | 46.9 | 30.8 |
| C.E. | 90 | ✗ | 53.5 | 81.3 | 58.8 | 38.2 | 58.2 | 41.2 | 33.3 | 54.7 | 35.2 |
| **LOOK (Ours)** | 90 | | 55.2 | 81.6 | 61.3 | 38.4 | 58.3 | 41.6 | 33.6 | 54.9 | 35.7 |
| SimCLR | 200 | | 55.5 | 81.8 | 61.4 | 37.9 | 57.7 | 40.9 | 33.3 | 54.6 | 35.3 |
| BYOL | 200 | | 55.3 | 81.4 | 61.1 | 37.9 | 57.8 | 40.9 | 33.2 | 54.3 | 35.0 |
| SwAV | 200 | ✓ | 55.4 | 81.5 | 61.4 | 37.6 | 57.6 | 40.3 | 33.1 | 54.2 | 35.1 |
| MoCo-v2 | 200 | | **57.0** | 82.3 | 63.3 | **39.2** | 58.8 | **42.5** | 34.3 | 55.5 | 36.6 |
| SimSiam | 200 | | **57.0** | **82.4** | **63.7** | **39.2** | **59.3** | 42.1 | **34.4** | **56.0** | **36.7** |
| SimSiam | 100 | | 54.3 | 80.0 | 60.0 | 35.8 | 54.4 | 38.5 | 31.4 | 51.4 | 33.5 |
| SupCon | 90 | ✓ | 55.3 | **82.3** | 61.5 | 38.9 | **59.0** | 41.7 | 33.9 | 55.4 | 36.1 |
| MoCo-v2 | 90 | | 56.1 | 81.6 | 62.4 | 37.5 | 56.8 | 40.5 | 32.9 | 53.6 | 35.2 |
| **LOOK (Ours)** | 90 | | **56.3** | **82.3** | **62.7** | **39.2** | **59.0** | **42.1** | **34.3** | **55.9** | **36.2** |

# E TRANSFERRING TO DETECTION AND SEGMENTATION

## E.1 EXPERIMENTAL SETTINGS

In the main experiments, we evaluate the transferability of pre-training methods with the downstream fine-grained classification. In this section, we further evaluate with more downstream tasks, *i.e.* object detection and instance segmentation, where the pre-trained models serve as the backbone to extract feature maps. We follow MoCo (He et al., 2020) to conduct the experiments on PASCAL VOC (Everingham et al., 2010) and COCO (Lin et al., 2014) datasets. The details are as follows:

**PASCAL VOC Object Detection.** We use the detector of Mask-RCNN with C4 backbone (He et al., 2017) and an extra Batch Normalization (BN) Layer for fine-tuning. We follow the $1\times$ schedule implemented in Detectron (Girshick et al., 2018), with 24k iterations decaying at 18k and 22k iterations. The image size is $[480, 800]$ during training and $800$ during test. The model is trained on VOC 2007 trainval + 2012 train, and tested on VOC 2012 val.

**COCO Object Detection and Segmentation.** We use the same backbone as VOC, and follow the $1\times$ schedule with 9k iterations decaying at 6k and 8k iterations. The image size is $[640, 800]$ during training and $800$ during test. The model is trained on COCO 2017 train and tested on 2017 val.

## E.2 RESULTS OF DETECTION AND SEGMENTATION

We report the results of transferring to detection and segmentation in Table 10, with the COCO-style metric AP, $AP_{50}$ and $AP_{75}$. The metrics of detection and segmentation are marked as $AP^{bb}$ and $AP^{mk}$, respectively. We compared the proposed LOOK with supervised methods C.E. and SupCon (Khosla et al., 2020) and self-supervised methods SimCLR (Chen et al., 2020a), BYOL (Grill et al., 2020), SwAV (Caron et al., 2020), MoCo-v2 (Chen et al., 2020b) and SimSiam (Chen & He, 2021). We notice that the way of training with two augmented views contribute to better transferring results, due to the higher locality-sensitive (further visualization analysis is shown in the following subsection). Thus, we develop an improved version of LOOK with two augmented views for training. Table 10 shows that the proposed LOOK performs comparable or better compared with existing pre-training methods. We also notice that another important factor is the epochs of the pre-training stage. Under similar pre-training epochs, LOOK could outperform the compared methods. In spite of the results, we observe that the transferability on locality-sensitive tasks, *i.e.* object detection and instance segmentation, are different from that on fine-grained classification task. The proposed LOOK under supervised settings is designed to capture more high-level semantic information and concentrate on the key object of the input image, thus works better for classification task. While the self-supervised methods will concentrate on more details of the whole image including the visual background, and works better for detection and segmentation tasks. We present additional analy-

sis based on visualization of the attention maps, and leave to the future work for deeper studies on improving the pre-training methods with varying transferability.

### E.3 VISUALIZATION OF ATTENTION MAP

For further analysis on the transferability of the compared pre-training methods, we follow DetCo (Xie et al., 2021) to visualize the attention map generated from the backbone models. We reshape the input image as $448 \times 448$ for better visualization. To compute the attention map, we calculate the average values of the output feature map in $14 \times 14$ along the feature channel, normalize it into $[0, 1]$, reshape the map into $448 \times 448$ with bi-linear interpolation and project it to the original image. We show the attention maps of three representative methods, *i.e.* SupCon, the proposed LOOK and MoCo-v2. Since the methods are all based on contrastive learning with similar range of output values, we could fairly compare the difference of them on attention maps.

Figure 9 and 10 show the visualization results on ImageNet and COCO, respectively. We show images of complex scene with more objects to better analyze for detection and segmentation transferring. Compared with SupCon as supervised methods, LOOK could concentrate on more semantic details besides the core area of images for its relaxed restriction of classification, *e.g.* objects that do not belong to the image labels. While SupCon will neglect these details without direct relation to labels, *e.g.* persons interacting with the labeled object. Compared with MoCo-v2 as self-supervised methods, though LOOK also concentrates on additional details in the image, the attention of MoCo-v2 is distributed more sparsely on the whole image including the background, which helps serve better for detection and segmentation tasks. Based on the visualization analysis, we conclude that supervised methods mainly concentrate on the objects highly-related to the image labels, which may harm the transferability to locality-sensitive tasks. Though LOOK could preserve more semantic details, there still leave space to both capture high-level global semantics and obvious local semantics for more powerful transferability.

## F ADDITIONAL ABLATION STUDIES

In the ablation study of Section 4.3, we show the results of varying queue size, momentum and number of neighbors. In this section, we further show the effect of the hyper-parameter temperature $\tau$ to LOOK. We maintain the decaying factor of temperature as $0.1$ and set different beginning value of $\tau$. The results are shown in Figure 8. It is observed that LOOK shows good robustness to the varying temperature, which attributes to that the direct filtering way of kNN provides a more stable positive sample set for learning.

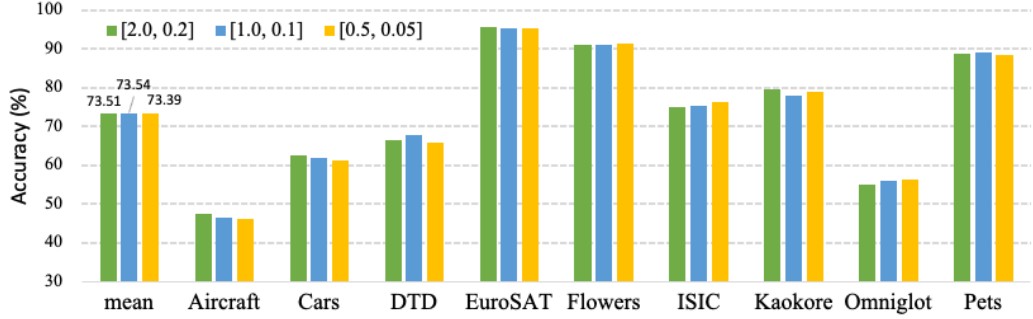

Figure 8: Linear fine-tuning results with different temperatures, where $[1.0, 0.1]$ indicates the temperature decaying from $1.0$ to $0.1$ linearly during the training stage.

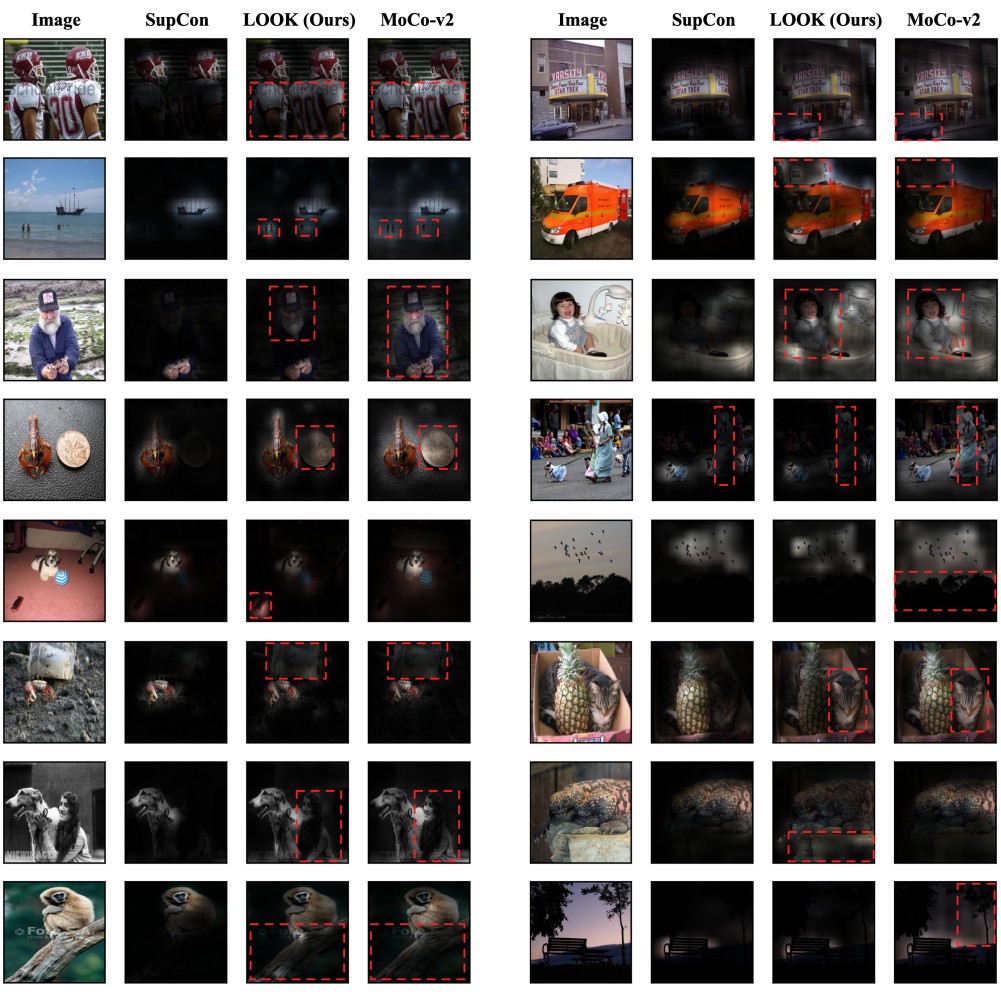

Figure 9: **Visualization of attention maps on ImageNet.**

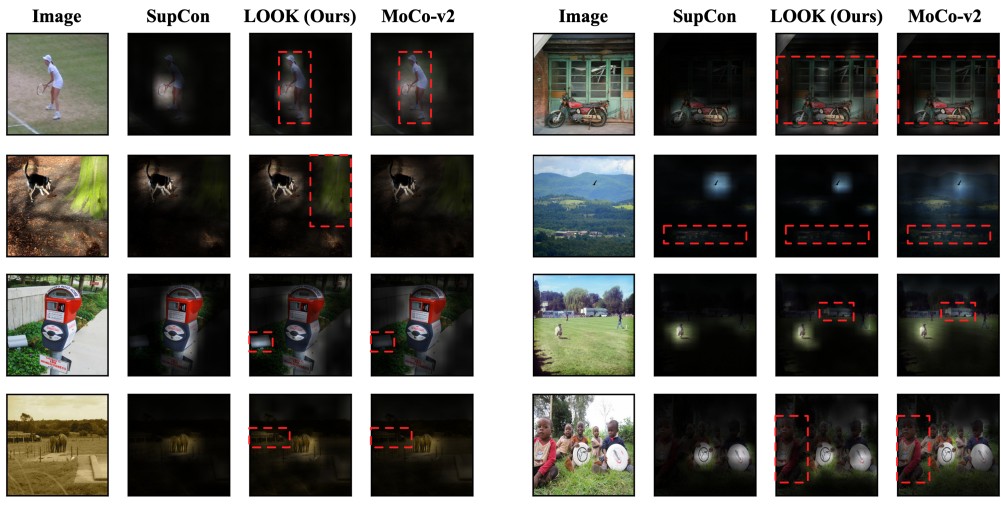

Figure 10: **Visualization of attention maps on COCO.**

