# OpenReview forum: "Rethinking Supervised Pre-Training for Better Downstream Transferring"
_ICLR.cc/2022/Conference — ICLR 2022 Poster_

### Official Review · Reviewer_mRy7 · 2021-10-28

**Correctness:** 2
**Technical Novelty And Significance:** 2
**Empirical Novelty And Significance:** 2
**Recommendation:** 6
**Confidence:** 4

**Main Review:**

The paper starts with a good observation that ImageNet categories often exhibit even finer categorical structures, such a static helmet in display and a helmet in action. These labels may be beneficial to the upstream pretraining task, however, it is potentially harmful for the downstream applications. Inspired by the observation, the paper seeks to find sub-clusters within a close neighborhood of each training sample. In this way, not all samples belong to the same category would collapse to a single point, thus allowing for sub-clusters.

While being pleased with the motivations, I have several concerns regarding to the details for the proposed approach.

1) I assume the data is randomly shuffled for each epoch. The paper uses a memory queue of 65536 samples in 1000 classes. Since ImageNet classes are well balanced, this means each class contains 65.5 samples in the queue on average. The paper uses a value of K=100 for k nearest neighbors, I wondering how many of the 65.5 samples fall in and out of the k=100 nearest neighbors. If all positives fall in the k nearest neighbors, there will not be sub-clusters to be emerged. Moreover, in what cases should a negative sample fall closer to the query sample, than some intra-class positives?

2) Following the previous comment, the ImageNet dataset contains 1300 image instances per category, which is a lot larger than 65.5 examples in the queue. I am wondering to what extent could 65.5/1300 (approximately 5%) amount of data represent a meaningful class. This is also related to how many sub-clusters usually occur within a class?

3) I appreciate the downstream evaluations for a set of classification tasks. It would be great to include the base performance on ImageNet with the proposed LOOK learning methods. Also, It would be also valuable to include the results on detection/segmentation transfer on VOC and MSCOCO. How is the performance when compared with contrastive or supervised baselines?

4) The leave-one-out nearest neighbor classification has been studied long ago, "Neighbourhood Components Analysis https://www.cs.toronto.edu/~hinton/absps/nca.pdf". The paper should be cited and discussed thoroughly.

**Summary Of The Paper:**

The paper proposes a new supervised method for visual pretraining by leveraging human labels. To avoid the intra-class instances to collapse into a single vector, the method implicitly encourages sub-clusters to occur within a category by a leave-one-out k-nearest neighbor classification loss. Compared with contrastive learning and several supervised baselines, the approach shows improved performance on several downstream tasks on car/flower/pets classifications, etc.

**Summary Of The Review:**

I am concerned about what actually sub-clusters are emerged from the LOOK model. Based on the current algorithm description, I believe that any positive examples even if they do not belong to the current sub-cluster would be sampled and drawn closer to the query sample, and hence no sub-cluster could be found. I guess lots of nearest neighbor visualizations would help answer this question.

---- Post rebuttal ----
I am satisfied by the additional analysis provided by the authors. I raised my rating to accept this paper.

---

> ### Author Response · Authors · 2021-11-13
> **Response to Reviewer mRy7 (3/3)**
>
> ***
> **[Concern 3]** Performance of LOOK on upstream ImageNet and detection/segmentation on VOC/MSCOCO.
>
> The upstream performance on ImageNet has been presented in Appendix Section D of the revised version. LOOK achieves kNN accuracy of 72.8% and 73.2% with and without strong data augmentation, which is lower but comparable with C.E.'s linear classification accuracy (75.5%). The results are reasonable since we sacrifice some upstream performance via relaxing the tightness of each class to achieve better downstream transferring. We also compare the upstream and downstream results of different pre-training methods, and observe that the downstream performance of one method is not highly related to its upstream performance. As we have discussed in the motivation of LOOK, methods with better upstream performance may fall in the over-fitting to the upstream datasets, leading to worse transferability on downstream tasks.
>
> For the detection and segmentation performance on VOC and MSCOCO, we have conducted the experiments with additional attention map analysis in the Section E of the revised version.
> ***
>
> **[Concern 4]** Citation and discussion on Neighborhood Components Analysis (NCA)
>
> Thanks for the reminding. In related works including Neighborhood Components Analysis (NCA) [3], SNCA (mini-batch version of NCA) [4], Examplar-v2 [2] and SupCon [1], each sample will be pulled togather with all samples in the dataset or memory bank with the same class. In contrast, the proposed LOOK only pull positive samples within the kNN, which alleviates the problem of neglecting intra-class difference and thus significantly improves transferability for downstream tasks. The difference could be observed from the positive set in the loss function, where all the samples are involved in NCA/SupCon and only samples within kNN are involved in LOOK. In other words, kNN is used to filter and select postive samples for constrastive learning in LOOK. We have cited NCA and presented discussion in the revised version.
> ***
>
> Please let us know if there are leftover concerns and we would be glad to do our utmost to address them.
>
> **References:**
>
> [1] Khosla P, Teterwak P, Wang C, et al. Supervised Contrastive Learning[J]. Advances in Neural Information Processing Systems, 2020, 33.
> [2] Zhao N, Wu Z, Lau R W H, et al. What makes instance discrimination good for transfer learning?[J]. arXiv preprint arXiv:2006.06606, 2020.
> [3] Goldberger J, Hinton G E, Roweis S T, et al. Neighbourhood components analysis[C]//Advances in neural information processing systems. 2005: 513-520.
> [4] Wu, Zhirong, Alexei A. Efros, and Stella X. Yu. "Improving generalization via scalable neighborhood component analysis." Proceedings of the European Conference on Computer Vision (ECCV). 2018.

---

> ### Author Response · Authors · 2021-11-13
> **Response to Reviewer mRy7 (2/3)**
>
> ***
> **[Concern 2]** Could the number of samples in queue represent a meaningful class or sub-class? How many sub-clusters usually occur within a class?
>
> We have presented a detailed discussion on class/sub-class learning with restricted queue size in the Appendix Section C.3 of the revised version.
>
> **Problem 2.1: Number of samples for supporting one class/sub-class**
>
> We take a further study to the effect of queue size on the ability of sub-class discovery, which will be conducted from two perspectives, *i.e.* empirical study and probability estimation. The following study is based on the assumption that each class could be uniformly divided into several sub-classes, which only serves as a simple analysis and could be further discussed in future work.
>
> - **Empirical Study.** Based on the expected number of samples for each sub-class in the queue, we could give a empirically estimated number of sub-classes that could be effectively learned based on experimental results. We refer to a similar work, SupCon [1], that utilizes similar contrastive learning inside large mini-batch or memory bank for classification learning. From the Figure 4 of SupCon, the model could reach satisfying performance on ImageNet-1K with more than 2,000 samples in batch, which contains 0.15% of all the 1,300 images per category. Then empirically, there is a potential of capturing $\frac{5\%}{0.15\%} \approx 33$ sub-classes for the proposed method with the default queue size of 65,536. It is noted that different from parametric classification learning that representing classes with parameters, LOOK adapts the non-parametric contrastive learning that continuously and rapidly update the memory samples throughout the dataset, which leaves sufficient times for each sample to meet and interact with samples from the same class or sub-class.
>
> - **Probability Estimation.** We further calculate the probability that all the sub-classes are sampled with at least one image in the queue. Results in the following Table show that for less than 12 sub-classes, all the training samples are almost guaranteed to find samples in kNN with the same sub-class. It is noted that for contrastive-based methods, even one positive sample could serve for effective training, *e.g.* the another augmented view in self-supervised learning.
>
> | number of sub-classes   |   < 6  |    8   |   12   |   16   |   20   |   24   |   28   |   32   |
>  |-----------------------|:------:|:------:|:------:|:------:|:------:|:------:|:------:|:------:|
>  | Probability           | 0.9999 | 0.9991 | 0.9677 | 0.8252 | 0.6002 | 0.4088 | 0.2821 | 0.2026 |
>
> ***
>
> **Problem 2.2: Number of sub-classes within a class**
>
> Since the definition of a class or sub-class is not fixed based on the class granularity and focused semantics, it is an open question to count all the sub-classes given a set of samples without artificial rules. A rough estimation to the averaged number of sub-classes discovered by LOOK could be the reciprocal of the averaged falling ratio, *i.e.* $\frac{1}{0.714} \approx 1.4$. We further analyze the relation of LOOK's components to the sub-classes discovery:
>
> - **Granularity**: The granularity of discovered sub-classes is affected by two factors, *i.e.* the memory size and $k$ of LOOK. With larger memory bank storing enough samples to cover the whole class together with smaller $k$ to strictly control the neighbors of samples, we could reach more fine-grained sub-classes. And there might exist a trade-off between memory-efficient training and sub-class discovery.
> - **Semantics**: The rule of splitting sub-classes is based on the learned visual semantics, which is guided by the supervision of coarse labels. Along the training stage, the semantics captured and focused by the model determine the kNN structure for sub-class discovery.
>
> Since the memory size is positively related to the number of sub-classes, it is still an challenging problem how to efficiently utilize the memory space to model more intra-class variance. For the future works, it is important to explore how to adaptively arrange memories for classes with more sub-classes and maintain representative samples in the memory for better coverage of all sub-classes.

---

> ### Author Response · Authors · 2021-11-13
> **Response to Reviewer mRy7 (1/3)**
>
> We thank the reviewer for the careful reading and valuable feedback to our submission. We appreciate for the agreement to our motivation based on good observations. We hope our responce could solve your concerns.
> ***
>
> **[Concern 1]** How many positive samples fall in and out of kNN? In what cases should a negative sample fall closer to the query sample than some intra-class positives?
>
> Thanks for the insightful concern. We have presented a detailed analysis on positive and negative samples inside kNN in the Appendix Section C.1 and C.2 of the revised version. Below is a summary of the analysis.
>
> **Problem 1.1: Number of positives falling in kNN**
>
> We calculate the number of samples in the queue for each class, and the averaged number of positive samples falling in the kNN. The results are shown in Figure 5 of Section C.1. We refer to the ratio of positive samples falling in kNN as the "falling ratio". It is noted that though the number of positive samples in queue is smaller than k for searching neighbors, not all the positives are forced and optimized to fall in kNN due to the relaxed restriction of LOOK. Our observation and analysis on the falling ratio are as follows:
>
> Compared among different classes, an unbalanced distribution of the falling ratio is observed, which indicates that their intra-class variances are different. To further investigate the relation between the falling ratio and intra-class variance, we visualize randomly selected samples from several classes with different falling ratio in Figure 6. The top classes with almost all positives falling in kNN show very similar visual appearance among each other. While for classes with falling ratio of about 0.5, we could observe some obvious sub-classes, *e.g.* the yellow and green crickets. And for classes with lower falling ratio, the samples show more diversity, such as velvet used in different things. In conclusion, the falling ratio of LOOK could adapt to different intra-class variance for learning efficient representation.
>
> Compared among different training stages, for the earlier stage from epoch 10 to 40, an obvious increasing of the falling ratio is observed, where we use hyper-parameters with larger aggregation range to pull samples into the kNN in a sparse distributed representation space. While for the latter stage from epoch 40 to 90, the falling ratio increases much slower in spite of the continuously increasing accuracy of training (as shown in Figure 3). Such a phenomenon indicates that based on LOOK's relaxed restriction of kNN learning, improving the classification performance will not force all the positive samples into the kNN. Therefore, we could maintain the sub-clusters discovered during training.
> ***
>
> **Problem 1.2: Negatives ranked closer than some positives**
>
> From the observation in Figure 5, there will also remain some negative samples in the kNN, especially for those with lower falling ratio, where some negatives rank closer to the query sample than some positives. To better understand such cases, we illustrate one of them in Figure 7. As we have discussed in Figure 1, LOOK discovers two sub-classes for the class "football helmet" as the helmet itself and its usage in match, respectively. For the shown query image in training, we observe some negative samples ranked before the positive one, where the positive belongs to different sub-class from the query's and the negatives actually show more similar appearance to the query.
>
> Based on the Figure 5-7, we conclude that for classes with higher intra-class variance, there will be a lower falling ratio and more negative samples of kNN. However, such a case is not bad because these negatives with similar visual appearance could occupy the left space of kNN and help to avoid positives from different sub-classes falling into the center of kNN.

---

> > ### Comment · Reviewer_mRy7 · 2021-11-26
> > **comments for positive ratios**
> >
> > Instead of positive ratios, I think a more convincing experiment would be to monitor the number of intra-class instances that never fall inside kNN for a specific image instance. Then average the this number across all datasets. What I am worrying is that although you do not enforce all intra-class samples to be closer in an explicit way, this might happen due to the training process of random sampling. If you can show that, for example, 40% intra-class samples on average never occur in the kNN space, that would be awesome.
> >
> > Even this could not be a direct evidence of sub-classes because the samples could connected via a manifold. For example, "a" is in a KNN of "b" but not "c", and "c" is in kNN of "b" but not "a". I would say this is still a single cluster, although the positive ratio is certainly not 100%.

---

> > > ### Author Response · Authors · 2021-11-26
> > > **Response to positive ratios (Part 2/2)**
> > >
> > > ***
> > > We also monitor the suggested ratio of positives that never fall in the kNN. To achieve this, we firstly compute the conditional probability of a sampled memory sample that does not fall into the kNN of query. Suppose the query is $x_0$ and the memory samples is sorted based on the their distance to query, *i.e.* $x_1, ..., x_r, ..., x_N$, where $N$ is the number of all samples. We denote the sampling set as $Q$ with size $q$, then the required conditional probability is
> > >
> > > $P(x_r \notin N_k(x_0)  |  x_r \in Q) = 1 - \Sigma_{i=0}^{k-1} \frac{C(N-r, q-i-1) *C(r-1, i)}{C(N-1, q-1)}$.
> > >
> > > We then calculate the real values based on the default settings of LOOK, and use the probabilities with specified thresholds to calculate the ratio of positives that never fall into kNN with the probability larger than 0.90, 0.95 and 0.99. The averaged results of all classes and the selected classes in Figure 6 are listed as follows. We also show the falling ratio for reference.
> > >
> > > |               |  mean  | "lady-slipper" | "echidna" | "sombrero" | "cricket" | "paperknife" | "velvet" |
> > > |---------------|:------:|:--------------:|:---------:|:----------:|:---------:|:------------:|:--------:|
> > > | p > 0.90      | 68.11% |     10.64%     |   15.89%  |   74.69%   |   75.61%  |    94.99%    |  97.89%  |
> > > | p > 0.95      | 67.69% |     10.27%     |   15.48%  |   74.40%   |   75.06%  |    94.87%    |  97.84%  |
> > > | p > 0.99      | 66.97% |      9.74%     |   14.86%  |   73.85%   |   74.20%  |    94.71%    |  97.78%  |
> > > | falling ratio |  0.71  |      1.00      |    0.98   |    0.63    |    0.57   |     0.18     |   0.14   |
> > >
> > > The ratio of never falling into kNN is shown to be negatively related to the original falling ratio. We observe that for a specified training instance, the exist enough percentage of memory instances that will almostly not fall into its kNN set, thus the training instance will not be enforced to reach these instances. The results suggest that the proposed LOOK can effectively control the learning process of training instances to only pull those neighbors with similar visual contents, which is consistent with the motivation of LOOK.
> > >
> > > ***
> > > Please let us know if there are leftover concerns and we would be glad to do our utmost to address them.

---

> > > ### Author Response · Authors · 2021-11-26
> > > **Response to positive ratios (Part 1/2)**
> > >
> > > Thanks for the valuable comments. First of all, we want to kindly claim that **the main purpose of LOOK is NOT to directly discover sub-classes in upstream dataset, but to improve downstream transferability** by filtering out some positive training pairs that do harm to downstream tasks. There are varying intra-class variance of different classes, and may not exist clear definitions or boundaries of sub-classes for all the classes. The proposed LOOK is designed to adapt to classes with different intra-class variance as follows:
> > >
> > > **Case 1: For classes with lower intra-class variance** (*e.g.* the echidna in Figure 6), more training positives will fall in the kNN set with higher falling ratios, then the proposed LOOK is similar to existing supervised methods like SupCon.
> > >
> > > **Case 2: For classes with clear sub-classes** (*e.g.* the green and yellow crickets in Figure 6) and lower falling ratios, LOOK works to avoid pulling samples from different sub-classes with the kNN filtering. Besides the studies in Section 4.5 and Appendix C.1, we present an additional analysis considering the indirect connections on manifold as suggested. Since the training samples and memory samples generate a bipartite graph based on kNN construction, we introduce the concept of multi-hop neighbors in graph learning and study the **multi-hop falling ratio**.
> > >
> > >  We calculate the $k$-hop falling ratio via the number of positives connecting to the query within no more than $k$ jumps in the bipartite graph. We monitor the 3-hop and 5-hop falling ratio for analysis. The observations and conclusions are as follows:
> > > - An unbalanced distribution of k-hop falling ratio is observed, which is similar to the original 1-hop falling ratio and further demonstrates the varying intra-class variance on ImageNet-1K classes.
> > > - The increasing speed of falling ratios from 1-hop to 5-hop also varies among classes. We list some randomly selected classes with different increasing speed in the following table. For classes with slower increasing of falling ratio, there exist two types:
> > >   - The original 1-hop falling ratio is high enough, which corresponds to case 1.
> > >   - The falling ratio maintain lower even with multi-hop, which suggests that there is a clear seperation of connected domains and thus sub-class distribution corresponding to case 2.
> > >
> > > We also visualize the images of the second type of classes and observe obvious sub-classes. The results could be included in the camera-ready version. We leave the classes with faster increasing falling ratios for the following discussion of case 3.
> > >
> > > |       |  #986 |  #25  |  #387 |  #88  |  #152 |  #222 |  #119 |
> > > |-------|:-----:|:-----:|:-----:|:-----:|:-----:|:-----:|:-----:|
> > >  | 1-hop | 0.922 | 0.737 | 0.516 | 0.495 | 0.562 | 0.336 | 0.199 |
> > > | 3-hop | 0.943 | 0.786 | 0.585 | 0.571 | 0.704 | 0.562 | 0.405 |
> > >  | 5-hop | 0.947 | 0.801 | 0.615 | 0.594 | 0.714 | 0.594 | 0.491 |
> > >
> > > **Case 3: For classes with higher intra-class variance but no clear split of sub-classes** (*e.g.* the paperknife in Figure 6), the calculation of kNN in LOOK works to generate a similarity-based graph (or the referred “manifold”) without multiple connected domains or sub-classes seperation. Under such cases, there will not be a clear separation of sub-classes, but a sparser distribution inside the whole class (*e.g.* the class in gray of Figure 4). We also study the multi-hop falling ratio for this case, where the 5-hop falling ratio increases rapidly compared with the 1-hop falling ratio.
> > > - From the aggregation-based graph learning, the aggregation effect to $k$-hop neighbors decreases lower with larger $k$. Thus, LOOK could avoid a strong enforcement of pulling all the samples closer to each, but spend more efforts to pulling 1-hop neighbors. Such an optimization helps to generate sparser intra-distribution, which has also been shown in Section 4.5 with visualization and intra-class distance measuring.
> > > - Furthermore, we claim that pulling $k$-hop neighbors might not be harmful to representation learning. Since the samples for these classes are with more various appearance, it helps us to mine more high-level semantics under the same label information.

---

### Official Review · Reviewer_wg1q · 2021-11-02

**Correctness:** 4
**Technical Novelty And Significance:** 3
**Empirical Novelty And Significance:** 3
**Recommendation:** 6
**Confidence:** 4

**Main Review:**

Strengths:
1. This paper is well organized and easy to understand.
2. The idea of using a stronger classifier (eg. KNN classifier) for preserving intra-class semantic differences is reasonable and shows impressive performance.
3. The experiments and analysis are sufficient.

Weaknesses:
1. Current experiments only present results on downstream classification tasks. It’s not clear how the proposed method will perform on other tasks like detection or segmentation.
2. It shows in Table 6 that compared to previous works, higher intra-class distance and lower inter-class distance will bring better transferability. However, it’s not clear how actually intra-class and inter-class variance influence transferability. It would be better to have more analysis on this issue so that the advantages of the proposed method can be better illustrated.


**Summary Of The Paper:**

This paper proposes LOOK, a new supervised pre-training method which can maintain intra-class semantic differences for better transferability to downstream tasks. Specifically, LOOK proposes to use a KNN classifier instead of the simple linear classifier for the pre-training task. Experiments show that LOOK outperforms previous supervised and unsupervised pre-training methods.

**Summary Of The Review:**

This paper is generally well motivated and well written. The experiments have sufficiently validated the effectiveness of the proposed method on downstream classification tasks. My minor concern is the performance on other downstream tasks.

---

> ### Author Response · Authors · 2021-11-13
> **Response to Reviewer wg1q**
>
> We thank the reviewer for the careful reading and valuable feedback to our submission. We are encouraged that the reviewer found our work is well organized with reasonable ideas and sufficient experiments. We hope our responce could solve your concerns.
> ***
>
> **[Concern 1]** Performance on detection/segmentation.
>
> Thanks for the suggestion. We have conducted additional experiments of object detection and instance segmentation, and the results with further attention map analysis are included in the Appendix Section E of the revised version.
> ***
>
> **[Concern 2]** Relationship between intra-/inter- class distance and transferability.
>
> Thanks for the suggestion. We add more discussions about this part. There have been works [1, 2] showing that the **class separation metric negatively correlates with linear transfer accuracy**. Through the calculation formula in [1], larger intra-class distance and inter-class distance will contribute to lower class separation, which means better transferability. From another viewpoint, by increasing the intra-class variance, it can be shown that the complexity measurement (*e.g.* VC dimension) of the resulting representation will be significantly larger than that for small intra-class variance, making it possible to represent a richer set of functions, which further improves the transferability of model.
> ***
>
> Please let us know if there are leftover concerns and we would be glad to do our utmost to address them.
>
>
> **References:**
>
> [1] Kornblith S, Lee H, Chen T, et al. What's in a Loss Function for Image Classification?[J]. arXiv preprint arXiv:2010.16402, 2020.
> [2] Zhao N, Wu Z, Lau R W H, et al. What makes instance discrimination good for transfer learning?[J]. arXiv preprint arXiv:2006.06606, 2020.

---

> > ### Comment · Reviewer_wg1q · 2021-11-29
> > **How is "LOOK with two augmented views" implemented?**
> >
> > In Appendix Section E, It shows that the original LOOK performs worse than state-of-the-art self-supervied pretraining on detection and segmentation downstream tasks, but "LOOK with two augmented views" performs better. So, how is the modified LOOK implemented? Simply adding augmented training images or using siamese networks with additional losses? It's not clear where the improvement comes from.

---

> > > ### Author Response · Authors · 2021-11-29
> > > **Responces to Reviewer wg1q**
> > >
> > > Thanks for your valuable feedback. The version of LOOK with two augmented views (referred as "LOOK-2V") is implemented by introducing another augmented view of training samples and adding a pulling loss to LOOK between the two augmented views. Based on such implementation, we compare with two types of pre-training methods as follows:
> > >
> > > - **Supervised:** There are also pulling effect of two augmented views in existing supervised contrastive learning methods (*e.g.* SupCon/Examplar-v2). Thus, the only difference of LOOK-2V to these methods is the removed positive samples outside the kNN, and the results show better transferring ability of LOOK loss, *e.g.* 56.3% (LOOK-2V) *v.s.* 55.3% (SupCon) on PASCAL VOC.
> > > - **Self-Supervised:** Based on the visualization of attention map, we observe that the transferabilty of self-supervised methods to detection/segmentation may attribute to their sensitivity to more local areas. In contrast, supervised methods mainly concentrate on the labeled object with higher level of semantics, thus work better for fine-grained classification tasks. We further observe that introducing another augmented views with pulling loss could improve similar locality-sensitivity of LOOK, which helps us to achieve better transferring results of detection and segmentation. In spite of that, as we have discussed in the Appendix E, there still leave spaces for future works to further improve transferability of pre-training with both high-level semantics and locality-sensitivity, such as semi-supervised pre-training based on the advantages of supervised LOOK and exising self-supervised methods. It is noted that as a contrastive learning method, LOOK could be easily combined with those self-supervised methods under contrastive learning.
> > >
> > > ***
> > > We would include the implementation details in the camera-ready version. Please let us know if there are leftover concerns and we would be glad to do our utmost to address them.

---

> > > > ### Comment · Reviewer_wg1q · 2021-11-29
> > > > **Final Comments**
> > > >
> > > > Thanks for the authors' quick response.
> > > > It's interesting to see that the two-augmented-view strategy also improves the proposed supervised learning method LOOK, but this strategy has little connection with the main insight/contribution of this paper.
> > > > Nevertheless, I would choose to keep my rating unchanged.

---

> > > > > ### Author Response · Authors · 2021-11-29
> > > > > **Responces to Reviewer wg1q**
> > > > >
> > > > > Thanks for your comment. The additional experimental results and observations in the appendix could serve as foundations for future works on pre-training and fine-tuning. Thanks again for your time with careful review and insightful feedback to our paper.

---

### Official Review · Reviewer_9e8a · 2021-11-04

**Correctness:** 3
**Technical Novelty And Significance:** 2
**Empirical Novelty And Significance:** 1
**Recommendation:** 5
**Confidence:** 4

**Main Review:**

Overall, I do not think this paper convinces me the proposed method can actually improve the performance of downstream tasks. Intuitively, I think the motivation of the paper makes sense, we might want to relax the class definition and allow multiple clusters inside each class based on visual similarity, however, I do not think the proposed method achieves that:

1. My first concern is about experimental results. Looking at Table 2, all the numbers from baseline are significantly lower than what's in the original paper. For example, in Table 3 of the BYOL paper (Grill et al.), BYOL reaches 88.1 on Aircraft, Standard CE reaches 86.0, SimCLR reaches 87.6/88.1, but in Table 2 of this paper, BYOL is only 65.56, CE is 69.73 and SimCLR is 67.27. I found all numbers of baseline are lower than what's claimed in the original paper. Similarly, the baseline numbers of SupCon are also lower here than in the paper (Table 4 in Khosla et al.). Note that all baseline numbers are obtained on ResNet-50.  My experience in fine-tuning these datasets with standard CE is more aligned with what is in the BYOL and SupCon paper. I suspect the authors might not use the optimal hyperparameters. In my opinion, without reasonable baseline results, it is hard to conclude that one method is better than others.

2. My second concern is about the explanation. The intuition of the paper is that we need to relax the cluster of each class. In Table 6, the proposed method actually has higher intra-class similarity. In my understanding, this means a tighter cluster of each class, which contradicts what the paper claims. It also has high inter-class similarity. The paper claims clearer boundaries. I think higher inter-class similarity usually means fuzzier boundaries, maybe the authors can explain further on this claim. This might be due to the confusion of using 'distance' in Table 6's title and 'similarity' in the explanation. Also, I think to support the claim, intra-class variance might be a better measurement

3. Last but not least, the paper uses a lot of ideas from recent self-supervised papers, such as memory queue from MoCo and MLP from SimCLR, and adds kNN loss on top of that. I am fine with utilizing the ideas that work, as long as the original idea in this paper actually shows performance improvements. However, based on 1), I cannot say the proposed loss is actually better.

**Summary Of The Paper:**

The paper proposes a leave-one-out kNN supervised method for pre-training on large-scale datasets.  The paper claims such a method will benefit downstream tasks by NOT enforcing each class, whether visually similar or not, to cluster together. Instead, each sample just needs to be close to the nearest K neighbors.  In this way, the model will not be overfitted to pre-train datasets. To implement this method, the paper uses a loss of weighted soft kNN loss with cosine distance and softmax normalization. To solve the large search space and feature update problem of kNN, the paper uses ideas from MoCo, utilizing a memory queue as a feature bank to store features of previous N samples and First-in-first-out to update the memory queue. To solve the convergence issue, the paper uses ideas from SimCLR and utilizes MLP instead of the linear layer to project features into a more compact space.

The proposed is tested on a standard fine-tuning setting, using ImageNet as the up-stream pre-training dataset and multiple fine-grained datasets as down-stream fine-tuning datasets. The proposed method is compared with both the supervised and self-supervised methods,


**Summary Of The Review:**

See in the Main Review

--- Post-rebuttal
Overall, I am still on the fence about the paper. However I am not against accepting it.

---

> ### Author Response · Authors · 2021-11-13
> **Response to Reviewer 9e8a (2/2)**
>
> ***
> **[Concern 2]** Confusion between the higher intra-class similarity in Table 6 and the explanation of our method.
>
> We sincerely apologize the confusing description in the paragraph "Measurements of intra-class and inter-class similarity" of Section 4.5, where we incorrectly refer to the intra-class "distance" as intra-class "similarity". Table 6 shows the measurement $1 - cosine()$ among samples inside and outside classes. As shown in the caption of Table 6, the larger value suggests higher variance of intra-class and inter-class samples. Therefore, LOOK actually achieves larger intra-class distance, *i.e.* lower intra-class similarity and less tight distribution of each class, which is consistent with our claims of motivation. We thank the reviewer for pointing out the confusion, and have re-writen this paragraph with unified word "distance" for better understanding in the revised version.
>
> As suggested, we also calculate the intra-class variance to further investigate intra-class distribution, which is measured as averaged variance per channel of each class. We show the results in the following table. It is observed that the intra-class variance shows consistent trends with the intra-class distance, and the proposed LOOK achieves higher intra-class variance.
>
> | | C.E. | SupCon | MoCo-v2 | SimSiam | LOOK (Ours) |
> |-----------------------------|:------:|:------:|:-------:|:-------:|:-----------:|
> | intra-class distance | 0.2746 | 0.3284 | 0.4377 | 0.3980 | 0.5758 |
> | intra-class variance (1e-4) | 1.341 | 1.603 | 2.138 | 1.943 | 2.903 |
>
> Furthermore, to better understand the relation between intra-class variance and downstream transferability, we refered to existing works [4, 5] showing that the class separation metric negatively correlates with linear transfer accuracy. Through the calculation formula in [5], larger intra-class distance and inter-class distance will contribute to lower class separation, which means better transferability.
> ***
>
> **[Concern 3]** The usage of existing ideas, *e.g.* MoCo queue and predictor module.
>
> The adapted techniques from existing self-supervised learning are mainly for extending the proposed LOOK method to larger dataset in pratical. For the effectiveness of the proposed loss, based on the results in the above responces, the proposed LOOK is shown to outperform existing methods under different settings of evaluation (optimized on both train and validation sets).
> ***
>
> Please let us know if there are leftover concerns and we would be glad to do our utmost to address them.
>
> **References:**
>
> [1] Grill J B, Strub F, Altché F, et al. Bootstrap Your Own Latent: A new approach to self-supervised learning[C]//Neural Information Processing Systems. 2020.
> [2] Khosla P, Teterwak P, Wang C, et al. Supervised Contrastive Learning[J]. Advances in Neural Information Processing Systems, 2020, 33.
> [3] Chen T, Kornblith S, Norouzi M, et al. A simple framework for contrastive learning of visual representations[C]//International conference on machine learning. PMLR, 2020: 1597-1607.
> [4] Islam A, Chen C F, Panda R, et al. A Broad Study on the Transferability of Visual Representations with Contrastive Learning[J]. arXiv preprint arXiv:2103.13517, 2021.
> [5] Kornblith S, Lee H, Chen T, et al. What's in a Loss Function for Image Classification?[J]. arXiv preprint arXiv:2010.16402, 2020.

---

> > ### Comment · Reviewer_9e8a · 2021-11-29
> > **Thanks for the clarifications**
> >
> > Appreciate authors' efforts on addressing my concerns.
> >
> > For [concern2] and [concern3], I think the authors addressed them thoroughly.

---

> ### Author Response · Authors · 2021-11-13
> **Response to Reviewer 9e8a (1/2)**
>
> We thank the reviewer for the careful reading and valuable feedback to our submission. We hope our responce could solve your concerns.
>
> **[Concern 1]** The experimental results of compared methods (BYOL/SupCon) are misaligned with the original paper.
>
> We sincerely apologize for the incomplete description of our experimental settings for fine-tuning. The reason of the misalignment is that **the results of BYOL/SupCon are based on models trained on both the training and validation sets** (see Secton D.3 of [1] and Section B.8.1 of [3]), while **our results are based on models trained only on the training set**. We have added more clear description of our settings in the revised version. Under the same dataset split of train/val/test, we compare the difference as follows:
>
> | stages                              |      BYOL/SupCon Settings      |        Our Settings        |
> |-------------------------------------|:------------------------------|:--------------------------|
> | stage 1: search of hyper-parameters |    train (train), eval (val)   |  train (train), eval (val) |
> | stage 2: evaluation (for report)    | train (**train+val**), eval (test) | train (**train**), eval (test) |
>
> We conduct the evaluation under our settings of fine-tuning and hyper-parameter searching since there is no open-source code to reproduce their transferring results. We also follow the settings from existing works such as [4].
> We further show that **though with different absolute values of accuracy, whether training on "train+val" or "train" will not influence on the comparsion of different pre-training methods, and our settings are also valid for transferability evaluation.**
> In the following table, we list our results with training on "train+val" and evaluating on "test" at stage 2. It is shown that the values reach a comparable range with those in BYOL/SupCon, *e.g.* 80%+ on Aircraft, 90%+ on Cars and 70%+ on DTD. The comparison among methods is consistent with Table 2 in the manuscript, *e.g.* LOOK could outperform C.E./SupCon/SimSiam, which demonstrates the comparison results under our settings also work for the evaluation of transferability. We are still working to get results of all the compared methods.
>
> | method      | aug++ |  mean | Aircraft |  Cars |  DTD  | EuroSAT | Flowers |  ISIC | Kaokore | Omniglot |  Pets |
> |-------------|:-----:|:-----:|:--------:|:-----:|:-----:|:-------:|:-------:|:-----:|:-------:|:--------:|:-----:|
> | C.E.          |       | 87.54 |   82.65  | 88.99 | 72.39 |  98.41  |  95.66  | 79.97 |  87.82  |   89.48  | 92.45 |
> | SupCon      |   √   | 86.91 |   81.91  | 89.23 | 71.33 |  98.91  |  94.42  | 76.77 |  88.43  |   90.69  | 90.49 |
> | SimSiam     |   √   | 87.06 |   86.05  | 90.29 | 68.78 |  98.52  |  95.43  | 76.17 |  88.92  |   92.99  | 86.37 |
> | LOOK (Ours) |       | **88.35** |   84.13  | 90.13 | 73.72 |  98.74  |  96.57  | 81.20 |  87.33  |   90.77  | 92.53 |
>
> We have conducted hyper-parameters searching of learning rate, batch size and weight decay, as shown in Section 4.1. It is noted that there is still a small gap of about 3% accuracy to the results of C.E. in BYOL/SupCon, which is attributed to their longer training steps (20k iterations of batch-size 256) and larger hyper-parameters grid-search space (7 learning rates X 8 weight decays). However, such a strategy is time-consuming. Evaluating one model on all the datasets in Table 2 will take about 11 days with 8 Tesla V100 GPUs. Thus, we adapt a faster fine-tuning and hyper-parameters searching strategy in [4], which is conducted to compare the transferability of different pre-training methods. (Datasets of [4] are only split into train/test, so the value of results are different from ours.) In spite of the computation cost, we are still trying our best to reproduce the fine-tuning results under BYOL/SupCon settings and compare with the proposed LOOK.

---

> > ### Comment · Reviewer_9e8a · 2021-11-29
> > **Thanks for explanations.**
> >
> > First of all, I appreciate authors' efforts on addressing my concern here. Now I am clear on the experiments setting and the reason of performance gaps between published results.
> >
> > I think the explanation on the 3% difference makes sense. Exhautive search for hyperparameters is indeed time-consuming and energy intensive. I would appreciate if the authors can at least show on one dataset (Aircraft would be great) that the proposed LOOK method can also scale with better hyperparameters and outperform C.E.  and self-supervised methods. Given that this is the final day for discussion (sorry for the late reply), this is not a requirement.
> >
> > Overall, I am still on the fence about the paper. However I am not against accepting it.

---

> > > ### Author Response · Authors · 2021-11-30
> > > **Response to Reviewer 9e8a**
> > >
> > > Thanks for your feedback. We are glad to hear that your concerns have been addressed by our explanations. For the additional results of LOOK under better searching space of hyperparameters, we will do our utmost to report it. But as you have mentioned, we may not catch up the close time of the discussion period, but we could include the results in the camera-ready version.
> > >
> > > Thanks again for your careful review and valuable comments to help us improve our submission.

---

> ### Author Response · Authors · 2021-11-29
> **A Gentle Reminder of Feedbacks**
>
> Dear Reviewer 9e8a,
>
> Thanks again for your careful reading and valuable comments to improve our submission. We want to leave a gentle reminder due to the closing end time of the discussion period. We have tried to address all your concerns with detailed explanations and results, and revised the paper correspondingly. We would really appreciate a feedback to make sure the responces and revisions have addressed all your concerns, or whether there is leftover concern we can address.
>
> Sincerely
> Authors of Paper3117

---

### Official Review · Reviewer_PfhP · 2021-11-07

**Correctness:** 3
**Technical Novelty And Significance:** 2
**Empirical Novelty And Significance:** 3
**Recommendation:** 6
**Confidence:** 4

**Main Review:**

+ As hinted in the summary, the biggest realization in the paper is to focus on such methods for *pre-training*, and not just for training the current task. So it has some empirical novelty.
+ The paper did a good job in empirically studying the transferred representations. E.g., it covers linear/full fine-tuning cases, different percentage of labels for some datasets, ablation studies, hyper-parameter searches, generalization to more backbones, parametric vs. non-parametric classification, visualizations, quantitative measures to justify the claims made in the introduction about intra- vs. inter- class distances. A lot of such studies make the paper rich and strong in terms of experimentation.
+ Besides some minor typos (please get it proof-read by others), the paper is well written and well organized. The illustrations, visualizations, and stories are also very clear.

- Idea-wise, I do not think the approach of using non-parametric classification for modern visual tasks is novel. E.g.:
Wu, Zhirong, Alexei A. Efros, and Stella X. Yu. "Improving generalization via scalable neighborhood component analysis." Proceedings of the European Conference on Computer Vision (ECCV). 2018.
This is a follow-up work of memory bank paper that applies it to supervised classification (and discovery of sub-classes) which this paper is not citing.
- Related: I feel comparing against CE or SupCon is a bit weak. I am sure in the literature there are already quite a few works (like the one above) that explore non-parametric classification. Their transfer learning ability may also be very good.
- I am not sure the paper's approach is really "leave-one-out". It is using a memory bank to store the examples, so there is no explicit operation to leave the current example of interest out. So to me the title is not so appropriate.
- Regarding gradient explosion: I am not sure I understand it, especially why having an extreme-value filtering strategy can be effective here.
- I understand the focus is on transfer learning, but at least it is still good to present the supervised learning results on ImageNet here.

* How does the method perform without l2 normalization? I assume compared to normal cross entropy learning, one could still do multi-positive cross-entropy by having multiple prototypes per-class. An important difference here is it follows SSL methods and used InfoNCE loss with l2 normalization. So I am wondering how important it is in the pipeline? Is it fine to remove it?
* A related question is for studying the temperature in the InfoNCE loss, how important is this?

**Summary Of The Paper:**

The paper presents an interesting approach that expands the recent success of self-supervised pre-training with *instance discrimination* to supervised pre-training. The key insight is to have each class being represented not just by a single weight vector, but in a non-parametric fashion via KNN lookup from a MoCo memory bank. The paper avoids to compare the results for direct, supervised learning, but rather focuses on transfer learning to downstream tasks -- to me this has some empirical novelty. The transfer learning experiments are done extensively, including many settings for quantitative comparisons and qualitative visualizations.

**Summary Of The Review:**

Overall I am on the acceptance side of the paper. While non-parametric classification has ben studied in the literature, the paper presents an interesting angle of checking its effectiveness for down-stream transfer. The proposed approach is not only quite effective according to the paper's experiments, but also giving some insight about why the popular SSL methods based on instance discrimination has good transfer abilities. Some revision of the paper is surely required (on top of proof reading), but to me it is above the bar of acceptance.

---

> ### Author Response · Authors · 2021-11-13
> **Response to Reviewer PfhP (2/2)**
>
> ***
> **[Concern 5]** About the supervised learning results on ImageNet.
>
> Thanks for your advice. The upstream performance on ImageNet has been presented in Appendix Section D of the revised version. LOOK achieves kNN accuracy of 72.8% and 73.2% with and without strong data augmentation, respectively, which is lower but comparable with C.E.'s linear classification accuracy (75.5%). The results are reasonable since we sacrifice some upstream performance via relaxing the tightness of each class to achieve better downstream transferring. We also compare the upstream and downstream results of different pre-training methods, and observe that the downstream performance of one method is not highly related to its upstream performance. As we have discussed in the motivation of LOOK, methods with better upstream performance may fall in the over-fitting to the upstream datasets, leading to worse transferability on downstream tasks.
> ***
>
> **[Concern 6**] The importance of L2 normalization.
>
> The L2 normalization here is used to control the range of results measured between the training samples and the selected positive samples in memory, which makes it easier to set the value of related hyper-parameters, *e.g.* the temperature, and present a proper weight of pulling/pushing samples. In our experiments, when directly removing the L2 normalization, the model could not converge in training. We find out the reason is that with the output from the predictor's fully-connected layer, there exist some large value of pairwise dot production, which are then fed into the softmax function and ruin the aggregation weights of labels. Under such circumstance, the real positive samples may never get a proper weight to contribute to correct label, then the gradients will explode and be cut off based on the extreme value filtering strategy. Though it might help by adjusting the temperture or the initialization of predictor, it seems better to keep the L2 normalization in the current pipeline. The conclusion is similar to recent contrastive learning methods.
> ***
>
> **[Concern 7]** The importance of the temperature hyper-parameter.
>
> Thanks for your advice. The hyper-parameter temperature in our loss, similar to that in the InfoNCE loss, plays an important rule to control the aggregation range of positive samples. We have included additional ablation studies on the temperature hyper-parameter in the Section F of the revised version, which also suggests the robustness of LOOK to its configurations.
>
> ***
>
> **[Concern 8]** Some minor typos.
>
> Thanks for your advice. We have got the paper proof-read by others and improved the writing for better understanding.
> ***
>
> Please let us know if there are leftover concerns and we would be glad to do our utmost to address them.

---

> ### Author Response · Authors · 2021-11-13
> **Response to Reviewer PfhP (1/2)**
>
> We thank the reviewer for the careful reading and valuable feedback to our submission. We are encouraged that the reviewer found our work is empirically novel, well writen and organized with rich and strong experiments. We hope our responce could solve your concerns.
> ***
> **[Concern 1]** Difference with existing non-parametric classification methods.
>
> There do exist some non-parametric classification methods based on different loss function and memory banks, including the refered SNCA (Improving generalization via scalable neighborhood component analysis) [1], SupCon [2] and Examplar [3]. We conclude the difference between these works and the proposed LOOK as follows.
> In these related works, each sample will be pulled together with all samples in the dataset or memory bank with the same class. In contrast, the proposed LOOK only pull positive samples within the kNN, which alleviates the problem of neglecting intra-class difference and thus improves transferability for downstream tasks. The difference could be observed from the positive set in the loss functions, where all the samples are involved in SNCA/SupCon and only samples within kNN are involved in LOOK. In other words, kNN is used to filter and select postive samples with similar visual appearance for revising the loss function in LOOK. While in the compared methods, kNN is only used during the evaluation stage with non-parametric classifier. We have cited SNCA and presented the discussion of difference in the revised version.
> ***
>
> **[Concern 2]** More non-parametric methods should be compared.
>
> Existing non-parametric methods includes two types, *i.e.* supervised and self-supervised. For supervised methods, we have compared SupCon [2] and Examplar-v2 [3] in the literature, and are still working to request or reproduce the refered SNCA [1] pre-trained models. For self-supervised methods, apart from the compared representative methods in the submission, we also compare with the combination of SupCon and SSL, which shows competitive performance on the downstream datasets [4]. We are still searching for more non-parametric methods in the literature for comparison of their transferring performance.
> ***
>
> **[Concern 3]** The implementation of "Leave-One-Out".
>
> We sincerely apologize for the missing of detailed implementation of leave-one-out. We actually implement the leave-one-out mechanism implicitly based on the queue mechanism. For the proposed LOOK, to achieve a convenient fashion of leave-one-out that filters out the query sample from the memory queue, we implement as follows: Given each mini-batch as input, we compute the LOOK loss before updating the queue with current mini-batch samples. Since the update of queue follows the "First-In-First-Out" (FIFO) strategy and queue size is significantly smaller than the dataset size, it has been a long period since the last time that current training samples are pushed into the queue. Thus, the mini-batch samples are very possible to have been popped out of the queue in the earlier training iterations. Such a strategy is convenient for avoiding additional operations to filter out each training sample individually, which achieves satisfying performance in our experiments. Based on the experiment, it is also okay to conduct filtering by ID for each training sample and the results show no obvious changes. We have included the details of implementation in the revised version.
> ***
>
> **[Concern 4]** How the extreme-value filtering avoids gradient explosion?
>
> In the very early stage of training, due to the poor representation of model, it sometimes happens that few or even no positive samples are collected within kNN. Under such circumstance, we will get extremely low value of prediction and the loss function of LOOK will get a extremely high value as shown in equation (2). The gradient explosion under such cases can be eliminated through filtering out extremely low values of label prediction.
> ***
>
> **References:**
>
> [1] Wu, Zhirong, Alexei A. Efros, and Stella X. Yu. "Improving generalization via scalable neighborhood component analysis." Proceedings of the European Conference on Computer Vision (ECCV). 2018.
> [2] Khosla P, Teterwak P, Wang C, et al. Supervised Contrastive Learning[J]. Advances in Neural Information Processing Systems, 2020, 33.
> [3] Zhao N, Wu Z, Lau R W H, et al. What makes instance discrimination good for transfer learning?[J]. arXiv preprint arXiv:2006.06606, 2020.

---

> ### Author Response · Authors · 2021-11-29
> **A Gentle Reminder of Feedbacks**
>
> Dear Reviewer PfhP,
>
> Thanks again for your careful reading and valuable comments to improve our submission. We want to leave a gentle reminder due to the closing end time of the discussion period. We have tried to address all your concerns with detailed explanations and results, and revised the paper correspondingly. We would really appreciate a feedback to make sure the responces and revisions have addressed all your concerns, or whether there is leftover concern we can address.
>
> Sincerely
> Authors of Paper3117

---

### Author Response · Authors · 2021-11-21
**Summary of the Paper Revision**

We thank all the reviewers for their careful reading and valuable comments. Based on these suggestions, we have revised the paper as follows:

- **Rewriting the description of some experimental settings**, including the fine-tuning process and intra-class distance analysis for avoiding confusions (Section 4.1 and 4.5).
- **Additional discussion on the difference between the proposed LOOK and existing methds**, especially with the refered supervised non-parametric classification methods (Section B.2).
- **Additional analysis of the proposed LOOK pre-training**. We analyze the varying intra-class variance based on the ratio of positive and negative samples falling into kNN (Section C.1), and how the problem is addressed by LOOK with relaxed classification learning (Section C.2). We further discuss the ability of sub-class discovery (Section C.3).
- **Performance on upstream task**, which shows that we may sacrifice some upstream performance for better downstream transferring (Section D).
- **Performance on more downstream tasks**, *i.e.* objection detection and instance segmentation. We also present visualization analysis based on the attention map of compared methods (Section E).
- **Additional ablation study** on the hyper-parameter temperature, which further shows the robustness of LOOK to configurations (Section F).

We hope the revisions and responces could help address the concerns of reviewers. And we are glad to answer and discuss on leftover concerns.

---

### Author Response · Authors · 2021-11-30
**Summary of the Discussion**

Dear Chairs and Reviewers,

Hope this message finds your well.

With the closing of the discussion period, we present a brief summary of our discussion with the reviewers as an overview for reference.

First of all, we thank all the reviewers for their careful reading and valuable comments. We are encouraged that the reviewers found our paper is based on good observation (*R4*) and reasonable motivation (*R2*), emprically novel (*R1*), well written and organized (*R1*, *R3*) with strong and sufficient experiments (*R1*, *R3*). We are also glad to receive positive feedbacks from the reviewers that their concerns have been addressed by our responses and paper revision.
***

We summarize the main concerns of the reviewers with the corresponding paper revision as follows.

**[Additional analysis and experiment results]**

- **Additional analysis of the proposed LOOK pre-training** in *Section C*.
  - *Section C.1*: analysis on the ratio of positive samples falling into kNN with its relation to class variance.
  - *Figure 6*: visualization of samples from classes with different falling ratios and intra-class variance.
  - *Section C.2*: ranking of positive and negative samples optimized in LOOK.
  - *Section C.3*: analysis on the ability of sub-class discovery based on emprical and probabilty estimation.
- **Performance on upstream task** in *Table 9* of *Section D*.
  - *Table 9*: pre-training accuracy of compared methods on ImageNet.
- **Additional transferring results on detection/segmentation** in *Table 10*, *Figure 9 & 10* of *Section E*.
  - *Table 10*: transferring results of LOOK and compared pre-training methods on PASCAL VOC [1] and COCO [2].
  - *Figure 9 & 10*: visualization analysis on the attention map of compared methods.
- **Additional ablation study** on temperature hyperparameter in *Figure 8* of *Section F*.
  - *Figure 8*: linear fine-tuning results with different temperature values.

**[Revision of writings and descriptions]**
- *Section 4.1 & B.3*: clarifying some details of experimental settings for fine-tuning.
- *Section 4.5*: rewriting the analysis on intra-class distance with unified denotations to avoid confusion.
- *Section B.1*: adding more details of upstream pre-training
  - implementation details of *leave-one-out* of LOOK in practical.
  - discussion on the difference of LOOK with more supervised methods, *e.g.* SNCA [3].
- *Minor typos*: improving the paper with proof-read by others and fixing typos.
***

Based on the discussion with reviewers, we also present a brief summary of our paper as follows:
- **Observation**: We revisit supervised pre-training methods, and observe that their worse transferability compared with existing self-supervised methods are caused by the overfitting to upstream tasks. There may exist great intra-class variance in the man-made class definition, which are neglected by existing supervised methods.
- **Motivation**: We could alleviate the upstream overfit by avoid pulling samples with completely different visual appearance closer to each, and preserve more semantics representing intra-class variance for better transferring.
- **Method**: We propose a new supervised method named LOOK, which only requires each sample to share its label with most of its kNN samples. We develop efficient implementation of LOOK that scales well to large datasets.
- **Evaluation**: Transferring results to multiple downstream tasks show that the proposed LOOK outperforms existing supervised and self-supervised pre-training methods.
- **Future Works**: The study of LOOK could serve as foundations for further improving the downstream transferability of pre-training, connections between supervised and self-supervised methods to combine both advantages for stronger representation learning and transferring. Besides, based on the characteristics of efficient learning with relaxed label restriction, LOOK is also expected to be developed for scenarios with biased annotations, *e.g.* noisy labels and long-tailed distribution learning.

***

Thanks again for your efforts to the reviewing and discussion. And we appreciate all the valuable suggestions and feedbacks that help us to improve our submission.

Sincerely
Authors of Paper3117

**References**
[1] Everingham M, Van Gool L, Williams C K I, et al. The pascal visual object classes (voc) challenge[J]. International journal of computer vision, 2010, 88(2): 303-338.
[2] Lin T Y, Maire M, Belongie S, et al. Microsoft coco: Common objects in context[C]//European conference on computer vision. Springer, Cham, 2014: 740-755.
[3] Wu, Zhirong, Alexei A. Efros, and Stella X. Yu. "Improving generalization via scalable neighborhood component analysis." Proceedings of the European Conference on Computer Vision (ECCV). 2018.

---

### Decision · Program_Chairs · 2022-01-20

**Decision:**

Accept (Poster)

**Comment:**

The paper introduces a simple yet effective technique for supervised pre-training based on kNN lookup from a MoCo memory queue . Initially, the reviewers raised concerns about limited novelty with respect to neighborhood component analysis, baseline results lower than the original papers, and several other questions such as how many positive samples fall in and out of kNN. The author response was strong, adequately addressing the reviewer’s comments with additional experiments and clarifications. After the discussion period, three reviewers recommended borderline acceptance. One reviewer maintained score 5, suggesting a more exhaustive search for hyper-parameters, but indicated he/she was on the fence and would be ok if the paper is accepted. The AC considers the response of the authors regarding hyper-parameter search (and the small gap from other reported results) is reasonable, and agrees with the majority that the paper passes the acceptance bar of ICLR.